# SC-Arena: A Natural Language Benchmark for Single-Cell Reasoning with Knowledge-Augmented Evaluation

**Jiahao Zhao**[1,2,3]**, Feng Jiang**[2,†]**, Shaowei Qin**[2]**, Zhonghui Zhang**[2]**, Junhao Liu**[4]**,
Guibing Guo**[1,†]**, Hamid Alinejad-Rokny**[5]**, Min Yang**[2,3,†]

[1]Software College, Northeastern University, China
[2]Artificial Intelligence Research Institute,
Shenzhen University of Advanced Technology, Shenzhen, China
[3]Shenzhen Key Laboratory for High Performance Data Mining,
Shenzhen Institutes of Advanced Technology, Chinese Academy of Sciences, China
[4]University of California, Irvine, CA, USA
[5]School of Biomedical Engineering, UNSW Sydney, Sydney, Australia

`zhaojiahao@mails.neu.edu.cn, jiangfeng@suat-sz.edu.cn,`
`guogb@swc.neu.edu.cn, min.yang@siat.ac.cn`

## Abstract

Large language models (LLMs) are increasingly applied in scientific research, offering new capabilities for knowledge discovery and reasoning. In single-cell biology, however, evaluation practices for both general and specialized LLMs remain inadequate: existing benchmarks are fragmented across tasks, adopt formats such as multiple-choice classification that diverge from real-world usage, and rely on metrics lacking interpretability and biological grounding. We present **SC-ARENA**, a natural language evaluation framework tailored to single-cell foundation models. SC-ARENA formalizes a *virtual cell* abstraction that unifies evaluation targets by representing both intrinsic attributes and gene-level interactions. Within this paradigm, we define five natural language tasks (cell type annotation, captioning, generation, perturbation prediction, and scientific QA) that probe core reasoning capabilities in cellular biology. To overcome the limitations of brittle string-matching metrics, we introduce **knowledge-augmented evaluation**, which incorporates external ontologies, marker databases, and scientific literature to support biologically faithful and interpretable judgments. Experiments and analysis across both general-purpose and domain-specialized LLMs demonstrate that (i) under the *Virtual Cell* unified evaluation paradigm, current models achieve uneven performance on biologically complex tasks, particularly those demanding mechanistic or causal understanding; and (ii) our knowledge-augmented evaluation framework ensures biological correctness, provides interpretable, evidence-grounded rationales, and achieves high discriminative capacity, overcoming the brittleness and opacity of conventional metrics. SC-Arena thus provides a unified and interpretable framework for assessing LLMs in single-cell biology, pointing toward the development of biology-aligned, generalizable foundation models. Our code is available at: https://github.com/SUAT-AIRI/SC-Arena.

## 1 Introduction

Large language models (LLMs) are increasingly being applied in biological research, enabling knowledge extraction (Garcia et al., 2024), reasoning (Gong et al., 2023), and hypothesis generation (Abdel-Rehim et al., 2025) in diverse modalities. In cellular biology, researchers are actively inves-

---

[†]Corresponding authors.

tigating how to leverage LLMs to integrate high-dimensional molecular data with mechanistic understanding, which is crucial for tasks such as cell type annotation (Wu & Tang, 2025), perturbation analysis (Istrate et al., 2024), and mechanistic question-answering (Wang et al., 2024). Collectively, these efforts reflect the aspiration to construct a *virtual cell* (Roohani et al., 2025), a computational model that enables in silico analyses through simulations, thereby accelerating the scientific discovery process. However, realizing this vision requires the development of fair and comprehensive benchmarks that provide *interpretable*, *task-grounded*, and *biologically faithful* evaluations, capable of accurately assessing LLMs' ability to interpret biological signals and mimic single-cell behaviors beyond generic NLP metrics.

Existing benchmarks for comprehensively evaluating LLMs' ability to process heterogeneous single-cell biological data remain limited. Most focus on narrow tasks (e.g., cell type annotation (Yuan et al., 2024)) without assessing whether models acquire a holistic understanding of cellular identity and dynamics. Broader scientific QA benchmarks, such as SciBench (Wang et al., 2023) and PubMedQA (Jin et al., 2019), probe reasoning but remain domain-agnostic and fail to capture the demands of single-cell analysis. More recent efforts, such as CELLVERSE (Zhang et al., 2025) and SOAR (Liu et al., 2024), extend evaluation to multi-omics tasks, yet still lack systematic coverage of reasoning and generative capabilities. Consequently, a principled framework is still absent for determining whether LLMs can operate reliably across heterogeneous biological tasks while faithfully capturing biological attributes, dynamics, and reasoning.

Inspired by recent progress toward constructing *virtual cells*, which require models to account for cellular states (i.e., attributes) and generate corresponding responses (i.e., actions) to environmental cues, we introduce SC-ARENA, a benchmark that evaluates LLMs through the abstraction of a *Virtual Cell* within an arena-style evaluation setting. This paradigm reconceptualizes evaluation as a selection process: Can an LLM serve as a virtual cell by faithfully capturing biological attributes, dynamics, and reasoning? Concretely, we define minimal requirements for a virtual cell and design five representative natural language tasks: captioning, cell type annotation, cell generation, scientific QA, and perturbation prediction that jointly probe static properties and dynamic behaviors. In these tasks, unlike prior benchmarks that rely on constrained multiple-choice formats, we adopt open-ended QA to better reflect practical use cases and capture reasoning depth. Regarding evaluation, standard metrics such as accuracy or BLEU, widely used in previous works (Liu et al., 2024), cannot capture these aspects. We therefore adopt **LLM-as-a-judge** (Gu et al., 2024) while mitigating bias through a **knowledge-augmented framework** inspired by Eval-RAG Ryu et al. (2023) grounded in external databases and ontologies, resulting in evaluations that are interpretable, reproducible, and biologically faithful.

Experiments across multiple state-of-the-art LLMs demonstrate the utility of SC-ARENA. We find that (i) models perform well on text-aligned tasks such as captioning but struggle on perturbation prediction and mechanistic QA, revealing gaps in causal reasoning; (ii) knowledge-augmented evaluation correlates more strongly with expert judgments than string-based metrics; and (iii) even the strongest general-purpose LLMs fail to consistently simulate the attributes and methods of the Virtual Cell, underscoring the need for domain-specialized approaches. Our contributions are threefold:

**Virtual Cell abstraction.** We introduce the *Virtual Cell* as a unified evaluation object for single-cell reasoning, inspired by object-oriented modeling. This abstraction jointly encodes cellular *attributes* (identity, state) and *actions* (responses, interactions), enabling a principled and extensible framework for evaluation. Starting from five representative tasks, SC-ARENA systematically probes both static and dynamic aspects of cellular biology.

**Natural language and knowledge-augmented evaluation.** We reformulate single-cell benchmarks into natural language QA tasks, moving beyond rigid classification or multiple-choice formats to better reflect real-world usage. To ensure biological fidelity and interpretability, we design a knowledge-augmented evaluation scheme that integrates ontologies, marker databases, and literature evidence, providing domain-grounded scoring and explanatory rationales.

**Comprehensive empirical study.** We compare several popular general-purpose and single-cell specialized LLMs under SC-ARENA. The results show a strong alignment between our knowledge-augmented evaluator and expert judgments, while revealing systematic gaps, particularly in mechanistic reasoning, that highlight directions for future biology-aligned foundation models.

Table 1: Comparison of existing single-cell LLM evaluation frameworks.

| Framework | Evaluation Paradigm | Evaluation Format | Evaluation Metrics |
|---|---|---|---|
| **C2S-Scale** | Cell Sentence | Prompted Completion | Lexical & Statistical Matching |
| **Cell-o1** | Reasoning Agent | Constrained Selection | Classification Matching |
| **SOAR** | Single-Task Agent | QA-based Classification | Surface String Matching |
| **CELLVERSE** | Multi-omics Cell Sentence | Multiple Choice Questions | Classification Matching |
| **SC-ARENA (Ours)** | Virtual Cell | Open-Ended Natural Language QA | Knowledge-Grounded Matching |

## 2 RELATED WORK

### 2.1 SINGLE-CELL MODELING APPROACHES

The evolution of single-cell modeling has progressed from embedding-based architectures to natural language-driven reasoning. Early efforts applied Transformer architectures to large-scale single-cell RNA-seq corpora, encoding expression profiles into latent embeddings for downstream tasks. Geneformer (Theodoris et al., 2023) and scFoundation (Hao et al., 2024) trained transformer models from scratch on tens of millions of cells, whereas scBERT (Yang et al., 2022) and scGPT (Cui et al., 2024) adapted existing NLP architectures—BERT and GPT-2, respectively—for robust cell type annotation and perturbation prediction. CellFM (Zeng et al., 2025) further expanded model capacity, training 800M parameters on 100M cells to improve robustness and generalization. Beyond purely embedding-based paradigms, hybrid approaches such as scGenePT (Istrate et al., 2024) and Cellllama (Choi et al., 2024) integrated textual biological knowledge, marking an early transition toward language-driven modeling in single-cell research.

Building on the demonstrated effectiveness of general LLMs in cell annotation (e.g., GPTCell-type (Hou & Ji, 2024)), recent studies have reformulated single-cell modeling directly in natural language to enhance interpretability and reasoning. Cell2Sentence (Levine et al., 2024) and its extension Cell2Sentence-scale (Rizvi et al., 2025) introduced the concept of "cell sentences," converting gene expression profiles into textual representations and thereby laying the foundation for language-based modeling. More recent efforts, such as CellReasoner (Cao et al., 2025) and Cell-o1 (Fang et al., 2025), further emphasize reasoning, incorporating mechanisms for structured inference in single-cell tasks. However, as summarized in Table 1, these approaches operate under distinct paradigms that limit holistic assessment: C2S-Scale focuses on static encoding and scaling laws, evaluating prompted completion via lexical and statistical metrics (e.g., BERTScore, Gene Overlap); while Cell-o1 functions as a reasoning agent focusing on batch-level logical consistency, yet is constrained to selection-based formats (requiring candidate lists) rather than open-ended generation. Consequently, they lack a unified evaluation framework that systematically assesses biological reasoning capabilities through the lens of a dynamic Virtual Cell.

### 2.2 SINGLE-CELL BENCHMARKS

Several scientific QA benchmarks have been developed to evaluate LLMs in biomedical and STEM domains. General-purpose resources such as SciBench (Wang et al., 2023) and PubMedQA (Jin et al., 2019) adopt natural language QA formats and provide broad coverage of biomedical reasoning. However, they remain largely agnostic to single-cell contexts and lack the mechanistic depth required for cellular-level evaluation.

To enable more precise assessment in single-cell biology, specialized benchmarks have recently emerged. CELLVERSE (Zhang et al., 2025) focuses on unifying cross-modality data via multi-omics cell sentences, but relies on multiple-choice questions (MCQ), often converted from open queries to maintain stability, which constrains reasoning and does not reflect real-world usage. Similarly, SOAR (Liu et al., 2024) evaluates models as single-task agents focusing only on cell type an-

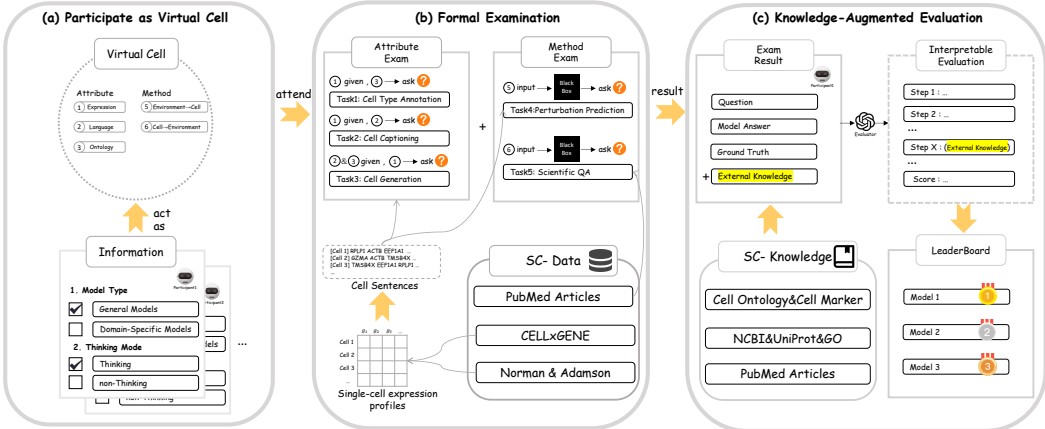

Figure 1: Overview of the SC-ARENA framework.

notation (targeting specific labels) and relies on string matching metrics (e.g., BLEU, exact match), reducing complex reasoning to surface-level lexical overlap and offering little interpretability. Crucially, as detailed in Table 1, these benchmarks utilize standard classification or string-matching metrics, lacking biological grounding in their evaluation methodology, and failing to assess whether models truly understand cellular mechanisms or merely memorize superficial patterns.

These limitations highlight the need for an evaluation framework that unifies assessment targets via a Virtual Cell (simulating attributes and methods), employs open-ended natural language reasoning without candidate lists, and incorporates domain-specific knowledge to ensure biological fidelity—a gap that our SC-ARENA framework addresses.

## 3 THE SC-ARENA EVALUATION FRAMEWORK

These limitations motivate the design of SC-ARENA, which builds on the *virtual cell* abstraction to unify evaluation targets and employs knowledge-grounded QA for interpretable and biologically faithful assessment, as shown in Figure 1. The benchmark consists of three components: (i) framing the participant model as a virtual cell, (ii) constructing a formal examination comprising five representative tasks, and (iii) applying our knowledge-augmented evaluation framework.

### 3.1 KNOWLEDGE CELL CLASS: DEFINING THE PARTICIPANT AS A VIRTUAL CELL

To evaluate whether LLMs acquire biologically grounded knowledge rather than memorizing superficial patterns, we introduce the notion of a *Virtual Cell*. A virtual cell serves as an abstraction of biological entities, defined as an instance of a *Knowledge Cell* class that encapsulates both static attributes and dynamic methods, which are defined as below:

**Attributes** Attributes represent the intrinsic identity and state of a cell, capturing multimodal biological information: (i) **Expression-based** features derived from scRNA-seq profiles, encoded as structured "cell sentences" (Rizvi et al., 2025), (ii) **Text-based** descriptions of morphology, function, localization, and role, curated from literature and databases, and (iii) **Ontology-based** hierarchical annotations from resources such as the Cell Ontology (CL).

**Methods** Methods represent the extrinsic dynamics of a cell, modeling its interactions with the environment: (i) **Cell** → **Environment** processes, including cytokine secretion, signaling, antigen presentation, and immune activation, and (ii) **Environment** → **Cell** responses, such as transcriptional changes under perturbations (e.g., drug treatment or gene knockout).

Leveraging this abstraction, a model that coherently represents both attributes and methods qualifies as a candidate **Virtual Cell LLM**, as it demonstrates the capacity to simulate both static identity

and dynamic behavior within a unified *Knowledge Cell* class. This design establishes a principled evaluation unit that integrates heterogeneous tasks into a single framework.

## 3.2 MULTI-TASK BENCHMARK WITH FORMAL EXAMINATION

Building on this definition, SC-ARENA is designed as a **multi-task benchmark** that operationalizes the evaluation of Virtual Cell LLMs across complementary perspectives. Each task probes a different mapping between modalities or reasoning direction within the *Knowledge Cell* class, and the tasks are defined as follows:

**Cell Type Annotation (Expression → Ontology)**: Assign ontology-grounded labels to expression profiles. Given a cell sentence, the LLM predicts the corresponding ontology-based cell type label.

**Cell Captioning (Expression → Language)**: Generate natural language descriptions from cell sentences. Given a cell sentence, the LLM produces a natural language description of the biological state, testing interpretability and the verbalization of transcriptomic patterns.

**Cell Generation (Ontology/Language → Expression)**: Synthesize plausible expression profiles from cell type descriptions or ontology terms. Given a cell type name, the LLM generates a plausible cell sentence, assessing its ability to produce molecular profiles consistent with semantic labels.

**Perturbation Prediction (Environment → Cell)**: Predict expression changes induced by perturbations given baseline profiles and perturbation signals. The evaluation requires the LLM to (i) predict up- and down-regulated genes and (ii) generate the post-perturbation cell sentence.

**Scientific QA (Cell → Environment)**: Answer mechanistic questions regarding cellular functions and intercellular interactions. Questions are derived from scientific literature, requiring the LLM to extract relevant knowledge from prior studies and provide evidence-based explanations.

Together, these tasks assess (i) bidirectional translation between molecular data and semantic descriptions, and (ii) reasoning over causal interactions between cells and their environments. By jointly covering static identity, dynamic behavior, and cross-modal reasoning, SC-ARENA provides a holistic testbed for measuring whether LLMs achieve a biologically meaningful understanding of cellular systems.

## 3.3 KNOWLEDGE-AUGMENTED EVALUATION

We found conventional NLP metrics (e.g., BLEU, ROUGE, BERTScore) fail to capture biological fidelity in our preliminary experiments (see Appendix A.4 for details). To address this, we introduce a **knowledge-augmented LLM-as-a-judge** framework, inspired by Eval-RAG (Ryu et al., 2023), which improves judging reliability by conditioning evaluation on retrieved context. Unlike conventional LLM-as-judge approaches (Gu et al., 2024) that rely only on the prompt and model output, our evaluator explicitly integrates curated external resources, including Cell Ontology, UniProt, Gene Ontology, CellMarker, and peer-reviewed literature. Grounding in these verifiable references enables the evaluation to capture semantic coherence, penalize biologically implausible outputs, and provide interpretable feedback. Formally, each evaluation instance is represented as

$$\mathcal{I} = (q, r, K, g),$$

where $q$ denotes the task prompt, $r$ the model response, $K$ the retrieved external knowledge, and $g$ the ground-truth answer. An evaluator LLM $E$ maps this tuple to a score

$$s = E(\mathcal{I}) \in [0, 100],$$

implemented as a discrete rating in $[0, 5]$ linearly rescaled to $[0, 100]$. Conditioning on both $K$ and $g$ allows the evaluator to accommodate linguistic variability, penalize factual errors against trusted references, and assign partial credit to semantically related predictions, yielding more faithful and interpretable evaluation than either string-matching metrics or ungrounded LLM-as-judge baselines.

Our framework prioritizes biological fidelity over speculative novelty: high scores are awarded only to predictions consistent with experimentally validated facts. This design ensures evaluation measures models' alignment with biological reality rather than their capacity for fluent but unsupported claims. The framework remains robust across different knowledge sources by anchoring judgments in curated, consensus-level evidence rather than dynamic retrieval, maintaining stability even when underlying databases (e.g., UniProt, NCBI, CellMarker) are substituted.

## 4 EXPERIMENTS

### 4.1 BENCHMARK DATASET CONSTRUCTION

All benchmark data sets are derived from publicly available high-quality single-cell resources. A detailed summary of these datasets is shown in Table 2.

Table 2: Summary of benchmark datasets in SC-ARENA.

| Task | Data Source | #Samples | Format (Input → Output) |
|------|-------------|----------|-------------------------|
| CTA | CELLxGENE | 608 | cell sentence (expression) → ontology label (CL) |
| CC | CELLxGENE | 608 | cell sentence (expression) → natural-language caption |
| CG | CELLxGENE | 608 | Ontology/cell-type name → cell sentence (expression) |
| PP | Norman; Adamson | 138 | Control & perturbed cell sentences + perturbation spec → (i) up/down DEGs; (ii) post-perturbation cell sentence |
| SQA | PubMed | 254 | Question → natural-language answer + evidence rationale |

The first three tasks—**Cell Type Annotation (CTA)**, **Cell Captioning (CC)**, and **Cell Generation (CG)**—are constructed from a shared subset of 608 representative profiles sampled from the *CZ CELLxGENE Discover* portal (Program et al., 2025). Each cell's gene expression profile is converted into a natural language "cell sentence," providing a unified representation across tasks. This design ensures consistency and comparability while establishing a closed-loop interplay among expression profiles, ontological labels, and natural language: models must identify cell identity, verbalize biological states, and generate plausible single-cell expression profiles.

**Perturbation Prediction (PP)** is compiled from two large-scale perturbation studies (Norman (Norman et al., 2019), Adamson (Adamson et al., 2016)), covering 138 genetic interventions. For each perturbation, we compute the mean expression profile of control (pre-perturbation) and perturbed (post-perturbation) cells, convert both into "cell sentences," and extract differentially expressed genes (DEGs) as ground-truth up/down-regulated gene sets.

Finally, **Scientific QA (SQA)** is curated from 100 PubMed articles focused on human genes and cellular biology, yielding 254 questions paired with reference answers and supporting evidence. Each question targets mechanistic reasoning, cell–environment interactions, and single-cell biology concepts, requiring application of the paper's findings rather than simple recall. Following semi-automatic pipelines such as EasyDataset (Miao et al., 2025), we streamline retrieval, question generation, and evidence linking, enabling interpretable evaluation of functional and mechanistic knowledge.

To evaluate the potential data leakage risk in our dataset, we conducted a verification using the C2S-scale series models in CTA, which were trained on source-related but differently formatted datasets. Despite this theoretical overlap, these models show significantly lower character-level similarity to our samples than general-purpose models, while achieving higher task accuracy. It indicates that the models did not memorize our data but instead learned task-relevant knowledge, suggesting that the leakage risk in our dataset construction is minimal. A detailed analysis is provided in Appendix A.6.

### 4.2 EXTERNAL KNOWLEDGE FOR EVALUATION

To ensure that evaluation reflects biological faithfulness rather than superficial lexical overlap, we ground each task in curated external resources:

**Cell Type Annotation:** We use hierarchical paths from the *Cell Ontology (CL)* (Diehl et al., 2016), which define standardized taxonomies of cell identity. By situating predicted types within this ontology, we can measure semantic similarity between model outputs and gold-standard labels, rewarding predictions that are close in the hierarchy even if not exact matches.

**Cell Captioning:** For evaluating natural language descriptions, we incorporate official definitions of target and ancestral cell types from CL. These definitions provide reference descriptions of mor-

phology, function, and localization, enabling assessment of whether generated captions capture the essential biological attributes and avoid omissions.

**Cell Generation:** We validate generated *cell sentences* against cell-type-specific marker genes curated in the *CellMarker* database (Zhang et al., 2019). Marker genes act as widely accepted gold standards for distinguishing cell identities, making them an ideal reference for judging whether synthetic profiles preserve biological distinctiveness.

**Perturbation Prediction:** To assess the plausibility of predicted differentially expressed genes (DEGs), we integrate functional annotations from *NCBI* (O'Leary et al., 2016), *UniProt* (UniProt Consortium, 2018), and the *Gene Ontology (GO)* (Carbon et al., 2021). These resources capture gene-level functions, pathways, and interactions, allowing us to verify whether predicted perturbation responses align with known biological mechanisms.

**Scientific QA:** For factual verification, we extract supporting abstracts and key excerpts from the original PubMed articles used to construct the questions. This provides ground-truth context for checking whether model answers are both scientifically accurate and evidence-supported.

## 4.3 EXPERIMENT SETUP

We evaluate both **general-purpose** and **domain-specialized** LLMs on SC-ARENA. General-purpose models include the **Qwen2.5** (Yang et al., 2024) and **Qwen3** (Yang et al., 2025) families across multiple scales, as well as **GPT-4o**, **DeepSeek-R1** (Guo et al., 2025), and **Kimi-K2** (Team et al., 2025). Domain-specialized models include **C2S-Scale** (Rizvi et al., 2025), **scGenePT** (Istrate et al., 2024), the reproduced **scGPT** (Cui et al., 2024), and the reasoning-oriented **Cell-O1** (Fang et al., 2025). Each domain-specific checkpoint is evaluated only on tasks aligned with its fine-tuning objective.

Each instance is standardized into a *cell sentence* representation. General-purpose LLMs receive a task-specific natural language query, while domain-specialized models follow their original inference protocols. All outputs are converted into a unified response schema before evaluation to ensure consistency. Additional details on the unified formatting, fairness controls, and cross-model standardization are provided in Appendix A.5.

Table 3: Performance of different models across five tasks (with Total Score). **Bold** = 1st place in column, Underline = 2nd place in column. Non-numeric entries ( — ) are excluded from ranking.

| Model | CTA | CG | CC | PP | SQA | Total |
|---|---|---|---|---|---|---|
| **General-purpose Models** | | | | | | |
| Qwen2.5-7B | 12.61 | 45.98 | 51.05 | 28.84 | 64.09 | 202.57 |
| Qwen2.5-14B | 25.89 | 51.74 | 56.05 | 34.78 | 66.37 | 236.06 |
| Qwen2.5-32B | 26.78 | 50.95 | 55.46 | 36.23 | 66.77 | 238.06 |
| Qwen3-8B | 21.45 | 50.53 | 57.20 | 37.39 | 72.83 | 254.45 |
| Qwen3-14B | 29.17 | 15.50 | 60.88 | 32.89 | 72.20 | 210.64 |
| Qwen3-32B | 31.35 | 55.39 | 62.69 | **37.54** | 65.03 | 252.00 |
| Qwen3-235B | 37.47 | 52.76 | 62.03 | 35.94 | **74.48** | 262.68 |
| GPT-4o | 36.29 | 59.70 | 63.02 | 37.24 | 67.56 | 263.81 |
| DeepSeek-R1 | **40.81** | 62.24 | 66.51 | 36.23 | 70.87 | 276.66 |
| Kimi-K2 | 40.00 | **63.04** | **67.89** | 37.10 | 69.13 | **277.16** |
| **Domain-specialized Models** | | | | | | |
| C2S-Pythia-410m(cell-type-prediction) | 47.34 | — | — | — | — | — |
| C2S-Pythia-410m(cell-generation) | — | 20.30 | — | — | — | — |
| C2S-Scale-Pythia-1b-pt | 41.68 | 18.55 | — | — | — | — |
| Cell-o1 | 34.11 | 43.91 | 67.89 | 24.20 | 64.09 | 234.20 |
| ScGPT | — | — | — | 21.55 | — | — |
| ScGenePT(NCBI+UniProt) | — | — | — | 24.13 | — | — |
| ScGenePT(GO-all) | — | — | — | 26.03 | — | — |

## 4.4 BENCHMARKING RESULTS

**Overall Performance.** As shown in Table 3, no system reaches the level of a reliable "virtual cell." Even the best-performing general models, Kimi-K2 (277.2) and DeepSeek-R1 (276.7), fall short of a normalized passing threshold (5*60). This highlights both the inherent difficulty of single-cell reasoning and the considerable headroom for improvement.

**Effect of Model Scale and Iteration.** Scaling and iteration bring consistent gains. Within the Qwen family, performance rises from 202.6 (Qwen2.5-7B) to 262.7 (Qwen3-235B), a nearly 60-point improvement. Iterative upgrades also matter: Qwen3 models outperform Qwen2.5 at comparable scales, and frontier systems like GPT-4o (263.8) and Kimi-K2 (277.2) surpass earlier baselines. However, no model is uniformly strong, Kimi-K2 excels in generation (63.0) and captioning (67.9), while Qwen3-32B narrowly leads in perturbation prediction (37.5). Scaling enhances fluency and coverage but does not resolve mechanistic reasoning. A task-level visualization using radar plots, along with further analysis, is provided in Appendix A.8.1.

**Task-wise Differences.** Performance varies sharply across tasks. Captioning (up to 67.9 with Kimi-K2) and scientific QA (74.5 with Qwen3-235B) reach the 60–70 range, while cell type annotation lags around 40 (best: DeepSeek-R1 at 40.8) and perturbation prediction remains below 38 (best: Qwen3-32B at 37.5). This asymmetry reflects the "fluent but not faithful" gap: models generate coherent text yet struggle with ontological precision and causal inference. A radar plot providing a task-level visualization of these disparities is shown in Appendix A.5.

**Domain-Specific vs. General Models.** Domain-specialized systems show complementary strengths. C2S-Pythia (410M, cell-type prediction) reaches 47.3 on annotation, outperforming even GPT-4o (36.3) and Qwen3-235B (37.5), despite being orders of magnitude smaller. By contrast, sc-GenePT variants achieve only 21–26 on perturbation prediction, far below general-purpose leaders, illustrating that specialization is highly task-dependent and not uniformly beneficial.

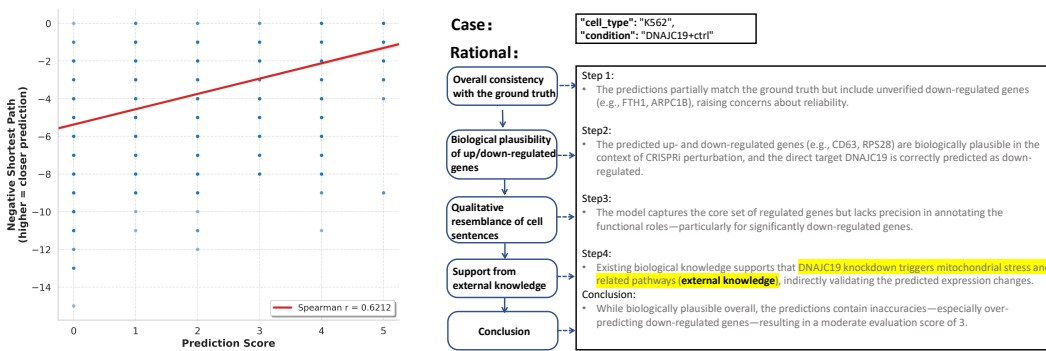

(a) The data points aggregated across all models in the CTA task.

(b) Example of scoring responses, produced by the evaluator using external knowledge in the PP task.

Figure 2: (a) Relationship between prediction score and ontology distance (Spearman $\rho = 0.6212$, p <0.001); (b) Example scoring responses using external knowledge.

## 5 ANALYSIS

### 5.1 BIOLOGICAL CORRECTNESS: ONTOLOGY-GROUNDED VALIDATION VIA CELL TYPE ANNOTATION

To assess whether the knowledge-augmented evaluator assigns biologically coherent scores, we analyze the task of cell type annotation, which naturally leverages the hierarchical structure of the Cell Ontology (CL). For each prediction, both the predicted and reference cell types are mapped to CL identifiers via the Ontology Lookup Service (OLS). We then compute their shortest-path distance $d_i$ within the CL hierarchy, using it as a proxy for biological relatedness.

We quantify alignment between evaluator scores and ontology distance by computing the Spearman rank correlation (Spearman, 1904; Kendall, 1970) between the evaluator's score $s_i$ and the *negative*

ontological distance $-d_i$, as shown in Figure 2a. The correlation is strongly positive ($\rho = 0.6212$, $p < 0.001$), indicating that predictions closer to the ground-truth type in the ontology consistently receive higher scores. This demonstrates that SC-ARENA's scoring scheme faithfully aligns with biological hierarchy, successfully operationalizing and validating the knowledge-augmented evaluation paradigm proposed in this work. A concrete example illustrating this scoring mechanism is provided in Appendix A.9. Beyond this CTA-focused analysis, we comprehensively validate the cross-task reliability of our knowledge-augmented evaluator across all five SC-ARENA tasks in Appendix A.7, demonstrating its consistent alignment with biological ground truth and expert judgment despite handling divergent output formats and reasoning demands.

## 5.2 Interpretability: Structured, Knowledge-Grounded Rationales

A central advantage of SC-ARENA's natural language–based design is its transparent and interpretable evaluation: each score is accompanied by structured rationales grounded in biological knowledge, rather than presented as an opaque number.

As shown in Figure 2b, in the perturbation prediction task the evaluator LLM generates biologically informed explanations that explicitly connect its scoring decisions to domain knowledge—covering gene function (e.g., *VIM* in stress response) and perturbation mechanism (e.g., *ARID1A* in chromatin remodeling).

This design transforms evaluation from a black-box judgment into an auditable and instructive process: it reveals *why* predictions succeed or fail, and turns evaluation into a teaching signal for iterative model refinement. It highlights that SC-ARENA not only measures performance but also explains it, enabling systematic error analysis and distinctive interpretability.

## 5.3 Discriminative Capacity: Distinguishing Biologically Meaningful Predictions

Beyond correctness and interpretability, an effective evaluation framework must also demonstrate discriminative capacity: the ability to distinguish models and outputs according to their biological plausibility. Traditional NLP metrics fall short in this regard.

For each task, we computed the similarity between model outputs and ground-truth references using several widely adopted NLP metrics, including BERTScore (Zhang et al., 2020), BLEU (Papineni et al., 2002), ROUGE (Lin, 2004), and METEOR (Banerjee & Lavie, 2005), with detailed results provided in Appendix A.4. However, the results reveal clear limitations: the scores are either uniformly close across models, offering little discriminative power, or near zero, failing to capture meaningful differences in biological reasoning quality.

In contrast, SC-ARENA achieves fine-grained discrimination by integrating structured rationales with domain knowledge to evaluate prediction plausibility and relative model strength. For example, in the cell type annotation task, SC-ARENA leverages cell ontology as external knowledge to capture differences in prediction depth. As shown in Appendix A.8.2, larger models tend to generate deeper, more specific cell type predictions, which align with their overall benchmark performance, enhancing the framework's discriminative power.

## 6 Discussion

### 6.1 Bridging the Gap: From Fluent to Faithful Biological Language Models

Our results in SC-ARENA reveal a clear dissociation between linguistic fluency and biological faithfulness in current LLMs. As shown in Table 3, general-purpose models consistently outperform domain-specialized ones on open-ended generation tasks such as Cell Captioning, demonstrating strong surface-level fluency. However, this advantage vanishes on tasks requiring ontological precision or causal accuracy: in Cell Type Annotation, most general models are outperformed by specialized counterparts, and performance on Perturbation Prediction remains universally poor across all models. Together, these findings expose a systemic "fluent but not faithful" gap: models may speak biology convincingly, yet fail to reason with the precision, hierarchy, and causality that define biological understanding.

To bridge this gap, our findings indicate that modeling must prioritize structured resources (e.g., ontologies, pathways) to encode biological logic. Regarding evaluation, SC-ARENA establishes that benchmarks must (1) assess granularity over exact matching, and (2) require auditable, knowledge-grounded rationales—principles operationalized in this work. Future integration with dynamic knowledge graphs will further shift models from merely *speaking* biology to truly *reasoning*.

### 6.2 SCORING RELIABILITY: ON THE CORRECTNESS OF LLM-AS-A-JUDGE

The second dimension concerns the reliability of scoring. Our knowledge-augmented LLM judge demonstrates measurable alignment with biological hierarchy: in the cell type annotation task, evaluator scores show a strong positive correlation with ontology distance, and the generated rationales explicitly reference domain knowledge such as gene functions and perturbation mechanisms. This demonstrates that the judge not only distinguishes biologically closer from more distant predictions but also grounds its decisions in interpretable reasoning, moving evaluation from opaque numbers to auditable explanations. Such capacity is critical for systematic error analysis, allowing evaluation to reveal not just *what* a model gets wrong, but also *why*.

Despite these strengths, the judge inherits the probabilistic nature of LLMs and thus exhibits limitations. Future improvements could mitigate these weaknesses in several ways: ensembling multiple judges to reduce variance across single models; calibrating against expert-annotated rationale sets to ensure that rationales reflect causal biological truth rather than spurious correlations; and integrating live biological knowledge bases such as GO, CL, and CellMarker so that scoring criteria evolve alongside scientific progress. Taken together, these advances could transform LLM-as-a-judge from a promising scaffold into a verifiable instrument for scientific evaluation. Further analyses in Appendix A.8.3 confirm that our evaluator is robust to the choice of judge model and invariant to response length, reinforcing its suitability as a reliable assessment tool.

### 6.3 FUTURE WORK: EXTENDING SC-ARENA TO EMERGING MODALITIES AND TEMPORAL REASONING

SC-ARENA is designed as a modular framework adaptable to evolving single-cell biology. Future extensions will incorporate: **(1) Spatial transcriptomics** for predicting cell-cell interactions within tissue context, extending the Virtual Cell's environmental dynamics; **(2) Developmental trajectories** to evaluate temporal reasoning through tasks like predicting progression in hematopoietic lineages; and **(3) Multi-omics integration** combining ATAC-seq, proteomics and other modalities to assess cross-modal biological consistency. These expansions will transform SC-ARENA into a living benchmark continuously aligned with emerging technologies and mechanistic questions in cell biology.

## 7 CONCLUSION

In this work, we present SC-ARENA, a natural language evaluation framework designed to assess the capabilities of foundation models on key single-cell biology tasks. By constructing a virtual cell abstraction and designing five representative tasks, including cell-type annotation, gene perturbation reasoning, and biological QA, we enable interpretable and task-grounded evaluation of LLMs in the biological domain. To enhance both precision and insight, we introduce *knowledge-augmented metrics* that leverage external databases to evaluate model outputs beyond surface correctness. Our experimental results reveal that current LLMs show promising yet uneven performance across tasks, and that knowledge grounding significantly improves evaluation reliability and interoperability. SC-ARENA provides not only a diagnostic tool for biological LLMs but also a new perspective on how natural language evaluation can be aligned with domain-specific reasoning. We hope this framework lays the groundwork for future efforts in building and benchmarking trustworthy, biology-aligned large language models.

### ACKNOWLEDGMENTS

This work was supported by the project of Shenzhen Application Research and Development Special Fund Support (Grant No. XLQSQ20250427092505008), the National Natural Science Foundation

of China (62376262, 62576083), the Natural Science Foundation of Guangdong Province of China (2024A1515030166, 2025B1515020032), Shenzhen Science and Technology Innovation Program (KQTD20190929172835662).

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

# A APPENDIX

## A.1 ETHICS STATEMENT

All data used in this work are derived from publicly available, open-source datasets, and thus raise no concerns regarding biomedical ethics or data licensing, as mentioned in the Section 4. We acknowledge that LLM-as-judge evaluation may still carry inherent biases in the discussion section.

Our contribution aims to mitigate such issues by explicitly grounding evaluation in external biological knowledge, providing a more objective framework and laying the foundation for more reliable and ethically sound evaluation practices in the future.

## A.2    REPRODUCIBILITY STATEMENT

Our experimental setup, including model selections, data preprocessing, prompt templates, inference protocols, and evaluation procedures, is fully described in Section 4.3 of the main text. Additional implementation details and evaluation protocols are provided in Appendix A.5. All benchmark datasets are constructed from publicly available resources as outlined in Section 4 and Table 2, ensuring full reproducibility of our results.

## A.3    THE USAGE OF THE LLM

In this paper, we only use LLM to polish the content to improve grammar and expression.

## A.4    THE DETAILS OF OTHER EVALUATION METRICS

Traditional NLP evaluation metrics, such as BERTScore, BLEU, ROUGE, and METEOR, are widely used for assessing model outputs. However, when applied to biological and domain-specific tasks, these metrics exhibit significant limitations. For instance, BERTScore often assigns nearly identical scores to outputs from different models, effectively collapsing biologically distinct predictions into similar numerical values. This is evident in Table 4, where models like Qwen3-8B, Qwen3-32B, and GPT-4o achieve comparable BERTScore values despite notable differences in the biological accuracy of their predictions.

Lexical overlap–based metrics, including BLEU and ROUGE, are equally problematic in this context. A model output such as "CD8 T cell, NK cell, B cell" can receive a high BLEU or ROUGE score against a gold label like "T cell" simply due to shared vocabulary, even though it is biologically incorrect. This can be observed in the perturbation task (Table 7), where BLEU-1 values are relatively high for several models, yet more detailed n-gram metrics (BLEU-2, ROUGE-2) remain low, reflecting partial but misleading lexical overlap rather than true biological fidelity.

METEOR, which accounts for synonymy and paraphrasing, provides slightly better differentiation, but it still lacks grounding in domain-specific knowledge and fails to penalize mechanistically implausible predictions. Across tasks such as cell type prediction, captioning, generation, perturbation, and ScienceQA (Tables 4–8), we consistently observe that high scores on these metrics do not necessarily correspond to biologically accurate or meaningful outputs. For example, DeepSeek-R1 often achieves the highest BLEU or METEOR scores in science QA and perturbation tasks, yet other models with slightly lower scores may produce more precise or mechanistically consistent predictions.

In summary, while these metrics provide a rough estimate of linguistic similarity, they are insufficient for evaluating the biological faithfulness of model outputs. Our observations underscore the need for specialized evaluation approaches that integrate domain knowledge and mechanistic constraints, rather than relying solely on traditional NLP metrics.

Table 4: Performance comparison of different models across traditional metrics in cell type annotation.

| Model | BERTScore | BLEU-1 | BLEU-2 | ROUGE-1 | ROUGE-2 | ROUGE-L | METEOR |
|---|---|---|---|---|---|---|---|
| Qwen2.5-7B | 80.20 | 0.24 | 0.02 | 0.12 | 0.02 | 0.12 | 0.07 |
| Qwen2.5-14B | 83.45 | 0.68 | 0.10 | 0.22 | 0.05 | 0.22 | 0.13 |
| Qwen2.5-32B | 82.93 | 0.94 | 0.08 | 0.23 | 0.04 | 0.23 | 0.07 |
| Qwen3-8B | 80.98 | 0.07 | 0.01 | 0.17 | 0.04 | 0.17 | 0.09 |
| Qwen3-14B | 80.86 | 18.38 | 4.54 | 0.21 | 0.06 | 0.21 | 0.12 |
| Qwen3-32B | 84.18 | 22.22 | 7.09 | 0.23 | 0.07 | 0.23 | 0.16 |
| Qwen3-235B | 83.47 | 11.52 | 3.04 | 0.29 | 0.10 | 0.29 | 0.18 |
| Kimi-K2 | 84.91 | 31.60 | 10.91 | 0.30 | 0.10 | 0.29 | 0.16 |
| GPT-4o | 84.11 | 11.49 | 2.15 | 0.28 | 0.09 | 0.28 | 0.12 |
| DeepSeek-R1 | 85.08 | 41.55 | 19.55 | 0.30 | 0.10 | 0.30 | 0.19 |

Table 5: Performance comparison of different models across traditional metrics in cell captioning.

| Model | BERTScore | BLEU-1 | BLEU-2 | ROUGE-1 | ROUGE-2 | ROUGE-L | METEOR |
|---|---|---|---|---|---|---|---|
| Qwen2.5-7B | 81.72 | 7.25 | 0.74 | 0.12 | 0.01 | 0.09 | 0.16 |
| Qwen2.5-14B | 82.29 | 8.44 | 0.89 | 0.13 | 0.01 | 0.09 | 0.17 |
| Qwen2.5-32B | 82.22 | 7.59 | 0.88 | 0.12 | 0.01 | 0.09 | 0.17 |
| Qwen3-8B | 82.16 | 3.49 | 0.32 | 0.12 | 0.01 | 0.09 | 0.15 |
| Qwen3-14B | 81.91 | 8.73 | 0.82 | 0.13 | 0.01 | 0.09 | 0.16 |
| Qwen3-32B | 81.87 | 8.07 | 0.73 | 0.13 | 0.01 | 0.09 | 0.16 |
| Qwen3-235B | 82.10 | 8.44 | 0.90 | 0.13 | 0.01 | 0.09 | 0.17 |
| Kimi-K2 | 81.47 | 8.50 | 0.93 | 0.13 | 0.02 | 0.09 | 0.16 |
| GPT-4o | 82.72 | 8.39 | 0.88 | 0.12 | 0.01 | 0.09 | 0.18 |
| DeepSeek-R1 | 82.18 | 8.77 | 1.04 | 0.13 | 0.02 | 0.10 | 0.18 |

Table 6: Performance comparison of different models across traditional metrics in cell generation.

| Model | BERTScore | BLEU-1 | BLEU-2 | ROUGE-1 | ROUGE-2 | ROUGE-L | METEOR |
|---|---|---|---|---|---|---|---|
| Qwen2.5-7B | 75.17 | 0.12 | 0.00 | 0.00 | 0.00 | 0.00 | 0.00 |
| Qwen2.5-14B | 77.01 | 0.36 | 0.00 | 0.01 | 0.00 | 0.01 | 0.00 |
| Qwen2.5-32B | 77.28 | 0.72 | 0.00 | 0.02 | 0.00 | 0.01 | 0.01 |
| Qwen3-8B | 75.17 | 0.08 | 0.00 | 0.00 | 0.00 | 0.00 | 0.00 |
| Qwen3-14B | 71.69 | 0.30 | 0.00 | 0.00 | 0.00 | 0.00 | 0.00 |
| Qwen3-32B | 77.27 | 0.33 | 0.00 | 0.01 | 0.00 | 0.01 | 0.01 |
| Qwen3-235B | 77.65 | 1.18 | 0.00 | 0.03 | 0.00 | 0.01 | 0.02 |
| Kimi-K2 | 79.24 | 1.91 | 0.00 | 0.04 | 0.00 | 0.02 | 0.02 |
| GPT-4o | 77.98 | 0.77 | 0.00 | 0.02 | 0.00 | 0.01 | 0.01 |
| DeepSeek-R1 | 79.60 | 2.67 | 0.00 | 0.06 | 0.00 | 0.02 | 0.03 |

Table 7: Performance comparison of different models across traditional metrics in perturbation prediction.

| Model | BERTScore | BLEU-1 | BLEU-2 | ROUGE-1 | ROUGE-2 | ROUGE-L | METEOR |
|---|---|---|---|---|---|---|---|
| Qwen2.5-7B | 87.40 | 88.39 | 29.92 | 0.56 | 0.08 | 0.21 | 0.34 |
| Qwen2.5-14B | 90.39 | 90.95 | 23.45 | 0.77 | 0.11 | 0.28 | 0.45 |
| Qwen2.5-32B | 90.92 | 91.99 | 22.01 | 0.83 | 0.12 | 0.30 | 0.48 |
| Qwen3-8B | 85.71 | 88.84 | 27.27 | 0.51 | 0.09 | 0.21 | 0.30 |
| Qwen3-14B | 88.19 | 86.11 | 24.98 | 0.62 | 0.10 | 0.24 | 0.37 |
| Qwen3-32B | 89.06 | 91.19 | 27.01 | 0.65 | 0.11 | 0.26 | 0.37 |
| Qwen3-235B | 89.20 | 90.19 | 25.55 | 0.70 | 0.10 | 0.26 | 0.42 |
| Kimi-K2 | 90.74 | 91.40 | 22.56 | 0.81 | 0.11 | 0.29 | 0.49 |
| GPT-4o | 90.50 | 91.53 | 18.26 | 0.84 | 0.12 | 0.30 | 0.48 |
| DeepSeek-R1 | 90.94 | 91.36 | 18.01 | 0.89 | 0.12 | 0.30 | 0.51 |

Table 8: Performance comparison of different models across traditional metrics in ScienceQA.

| Model | BERTScore | BLEU-1 | BLEU-2 | ROUGE-1 | ROUGE-2 | ROUGE-L | METEOR |
|---|---|---|---|---|---|---|---|
| Qwen2.5-7B | 88.51 | 43.22 | 14.10 | 0.38 | 0.13 | 0.23 | 0.25 |
| Qwen2.5-14B | 88.91 | 48.02 | 16.20 | 0.40 | 0.15 | 0.25 | 0.27 |
| Qwen2.5-32B | 89.04 | 52.43 | 17.42 | 0.40 | 0.15 | 0.25 | 0.26 |
| Qwen3-8B | 87.81 | 29.20 | 9.08 | 0.39 | 0.13 | 0.22 | 0.31 |
| Qwen3-14B | 85.46 | 32.65 | 9.73 | 0.39 | 0.12 | 0.22 | 0.30 |
| Qwen3-32B | 74.81 | 30.46 | 8.70 | 0.33 | 0.10 | 0.18 | 0.26 |
| Qwen3-235B | 87.55 | 29.86 | 8.66 | 0.39 | 0.12 | 0.22 | 0.30 |
| Kimi-K2 | 86.83 | 38.53 | 8.33 | 0.31 | 0.08 | 0.19 | 0.19 |
| GPT-4o | 89.24 | 53.73 | 18.48 | 0.40 | 0.15 | 0.25 | 0.26 |
| DeepSeek-R1 | 87.49 | 34.05 | 8.85 | 0.36 | 0.10 | 0.21 | 0.26 |

## A.5 EXPERIMENTAL DETAILS

Before detailing the models and our evaluation protocol, we provide a comprehensive overview of the data foundational to our benchmark. Table 2 summarizes the data sources, sample sizes, and specific input-output formats for all five tasks within SC-ARENA.

We evaluate a diverse set of both **general-purpose** and **domain-specialized** large language models (LLMs) to assess their performance across our single-cell biology benchmark suite. Our evaluation covers a range of publicly available foundation models, including **DeepSeek-R1** (Guo et al., 2025), **GPT-4o**, **Kimi-K2** (Team et al., 2025), and the **Qwen** series models, particularly **Qwen2.5** (Yang et al., 2024) and **Qwen3** (Yang et al., 2025). This selection spans a spectrum of model scales and architectures, allowing us to examine performance differences attributable to model capacity and pretraining strategies.

In addition, we include four LLMs that have been fine-tuned specifically for single-cell genomics: **scGPT** (Cui et al., 2024), **scGenePT** (Istrate et al., 2024), **C2S-Scale** (Rizvi et al., 2025), and **Cell-O1** (Fang et al., 2025). The **scGPT** model used in our evaluation was obtained from the version publicly released by the **scGenePT** authors; this model was fine-tuned on a perturbation dataset by the scGenePT team to produce the version used in our experiments.

Our evaluation proceeds in two stages: **answer generation** and **automated scoring**. Cell genes in each instance are first standardized into a unified cell sentence format. For general-purpose LLMs, each benchmark instance is reformulated as a natural language question using task-specific prompt templates (see Appendix for full prompt designs). The model receives this formatted input and generates a free-text response. For domain-specialized models, inputs are preprocessed according to each model's published inference protocol (e.g., tokenization and input schema), and outputs are post-processed into a unified response format compatible with our evaluation framework.

To eliminate evaluation artifacts arising from inconsistencies in model output conventions, all model responses, regardless of architecture or pretraining background, are converted into a standardized structured schema before being passed to the evaluator (e.g., `[Predicted_Cell_Type: ...]`, `[Up: geneA, geneB, ...]`). This ensures that all models are judged under identical formatting conditions, preventing discrepancies due to prompt conventions, tokenization, or decoding style.

At the input level, general-purpose LLMs uniformly adopt task-specific natural language prompts, whereas domain-specialized LLMs strictly follow their original, published inference protocols without any forced adaptation. This design prevents compatibility-induced biases and isolates the intrinsic reasoning or domain-specific competence of each model.

For transparency and interpretability, we explicitly categorize the evaluated systems into general-purpose versus domain-specialized models in the main paper. This separation reflects two meaningful axes of capability, general reasoning and biology-specific expertise, and avoids conflating these factors when interpreting performance differences.

We employ **GPT-4o-mini** as the automated evaluator for all tasks. For each task, a task-specific evaluation prompt is carefully designed. The evaluator is provided with the input question, the model-generated response, the ground-truth answer, relevant external knowledge, and the task-specific scor-

ing rubric (see Appendix for details). Based on this information, the evaluator assigns a score on a $[0, 5]$ scale. To facilitate cross-task comparison, these raw scores are normalized by dividing by 5, yielding a percentage score. The final task-level performance is obtained by averaging the normalized scores across all instances within the task.

To ensure the robustness of our results, each model was evaluated twice independently on every task. We confirmed that the discrepancy between repeated runs did not exceed 2 percentage points in accuracy, thereby validating the stability of our evaluation. These combined controls ensure that observed performance differences reflect genuine model capability rather than artifacts of training data similarity, input style alignment, or heterogeneous inference procedures.

The total score reported in Table 3 is the unweighted sum of the five task scores, reflecting the holistic nature of the Virtual Cell framework: each task—annotation, captioning, generation, perturbation prediction, and scientific QA—is considered an equally essential and complementary capability for modeling a complete virtual cell. We intentionally avoid arbitrary weighting to prevent introducing subjective biases, and instead present raw task-level performance transparently to allow users to interpret relative strengths according to their specific application needs.

### A.6 MITIGATING DATASET LEAKAGE IN SC-ARENA

Our evaluation paradigm is fundamentally different from prior works and is explicitly designed to minimize benchmark dataset leakage risks—especially pertinent given that models like C2S were trained on CELLxGENE data—through the following four strategies:

1. **New task formulations and prompts eliminate input–output overlap.** SC-ARENA introduces entirely new task definitions and natural-language prompts under the Virtual Cell abstraction. None of the input–output pairs, templates, or phrasings overlap with C2S or related training corpora. As a result, even if a model has seen the underlying raw scRNA-seq profiles, it has never encountered the linguistic formulation used in SC-ARENA.

2. **Evaluation is knowledge-grounded, not string-based.** Our LLM-as-a-judge assesses biological correctness using external resources (Cell Ontology, CellMarker, Gene Ontology, UniProt, and peer-reviewed literature), preventing models from gaining high scores through memorized patterns or lexical similarity. To further substantiate this, we computed BLEU scores for the two C2S models on the Cell Type Annotation (CTA) task (see Table 9). Both models achieve high biological accuracy despite remarkably low BLEU scores—lower than those of DeepSeek-R1, and Kimi-K2—strongly indicating that their performance reflects acquired biological knowledge rather than memorization of test samples.

3. **Randomized sampling reduces systematic overlap.** For datasets without predefined splits (e.g., CELLxGENE), we perform uniform random sampling across tissues, cell types, and expression distributions. This prevents overrepresentation of frequent or previously dominant classes and reduces the chance that performance arises from memorizing common patterns.

4. **SC-ARENA is a continuously evolving benchmark.** We acknowledge that any static benchmark may eventually become overfitted. SC-ARENA is therefore designed as a living benchmark: new perturbation datasets, multimodal atlases, and emerging biological tasks (e.g., Perturb-seq challenges) will be progressively incorporated under our Virtual Cell abstraction to further mitigate leakage concerns and enhance long-term robustness.

Table 9: BLEU scores for C2S models on the Cell Type Annotation (CTA) task. Low BLEU scores—despite strong biological performance—support that results are not due to data leakage or memorization.

| Model | BLEU-1 | BLEU-2 | CTA_Score |
|---|---|---|---|
| C2S-Pythia-410m | 23.73 | 17.14 | 47.34 |
| C2S-Scale-Pythia-1b | 13.66 | 8.29 | 41.68 |
| DeepSeek-R1 | 41.55 | 19.55 | 40.81 |
| Kimi-K2 | 31.60 | 10.91 | 40.00 |

## A.7 Cross-Task Reliability of the Knowledge-Augmented Evaluator

To address concerns about the generalizability of our knowledge-augmented evaluator beyond Cell Type Annotation (CTA), we conducted reliability analyses across all five SC-ARENA tasks, each involving distinct output formats and reasoning demands. As summarized in Table 10, we employ task-specific external validity metrics and expert judgments to validate the evaluator's alignment with biological truth and human assessment.

For **Cell Generation (CG)**, we measure the overlap between generated "cell sentences" and known cell-type-specific marker genes from CellMarker. The evaluator's scores show strong positive correlation with the percentage of correctly included markers (Spearman $\rho = 0.6669$, $p < 10^{-4}$).

For **Perturbation Prediction (PP)**, we compute cosine similarity between predicted and ground-truth sets of differentially expressed genes (DEGs). Evaluator scores correlate significantly with this similarity ($\rho = 0.6057$, $p < 10^{-138}$), confirming sensitivity to mechanistic fidelity.

For **Cell Captioning (CC)** and **Scientific QA (SQA)**, we conduct pairwise human preference studies: we randomly sampled 30 samples containing two responses from different models per task and ask two experts to independently judge which response is better. Crucially, the evaluator's rankings align well with aggregated expert preferences: $\rho = 0.8394$ ($p = 3.6 \times 10^{-9}$) for CC and $\rho = 0.8199$ ($p = 3.0 \times 10^{-8}$) for SQA.

These results collectively verify that our knowledge-augmented evaluator maintains high reliability across diverse reasoning modalities, structured prediction, molecular generation, and open-ended explanation, by grounding judgments in domain-specific knowledge and human expert alignment.

Table 10: Reliability validation of the knowledge-augmented evaluator across SC-ARENA tasks. Metrics reflect correlation between evaluator scores and external validity measures (Spearman $\rho$ and $p$-value). For open-ended tasks, we report expert agreement rate and correlation with evaluator rankings.

| Task | Validation Metric | Spearman $\rho$ | $p$-value |
|------|-------------------|-----------------|-----------|
| CTA | Ontology path distance | 0.6212 | $< 10^{-3}$ |
| CG | % correct CellMarker genes | 0.6669 | $< 10^{-3}$ |
| PP | DEG cosine similarity | 0.6057 | $< 10^{-3}$ |
| CC | Expert preference alignment | 0.8394 | $< 10^{-3}$ |
| SQA | Expert preference alignment | 0.8199 | $< 10^{-3}$ |

## A.8 Additional Reslut Analysis

### A.8.1 Additional Result Analysis with Radar Plots

**Task-level Visualization of Scaling Effects.** While the main text highlights numerical improvements from scaling and iteration, Radar plots provide a complementary, task-level perspective. Figure 3 compares representative general-purpose models across the five SC-ARENA tasks. The visualization confirms the aggregate trend reported in Table 3 — larger and newer models consistently expand the coverage of capabilities — but also exposes uneven gains across tasks. For instance, Kimi-K2 achieves a pronounced lead in captioning and generation, whereas Qwen3-32B performs comparatively better in perturbation prediction. These contrasts underscore that model scaling improves overall fluency and reasoning breadth, yet does not fully overcome the challenge of mechanistic prediction.

**Heterogeneity Across Tasks.** The radar plots further reveal that gains from scaling are not uniformly distributed. Open-ended tasks (captioning, scientific QA) show the steepest improvements, whereas deterministic tasks (cell type annotation, perturbation prediction) remain relatively constrained. This echoes the "fluent but not faithful" gap emphasized in the main discussion, illustrating how visualization helps to highlight task-specific limitations that may be obscured in aggregate scores.

**Implications for Model Development.** By making task asymmetries visible, radar plots emphasize the importance of fine-grained evaluation beyond single total scores. They suggest that future progress may require not only scaling and iteration, but also targeted approaches that explicitly address mechanistic reasoning in biology.

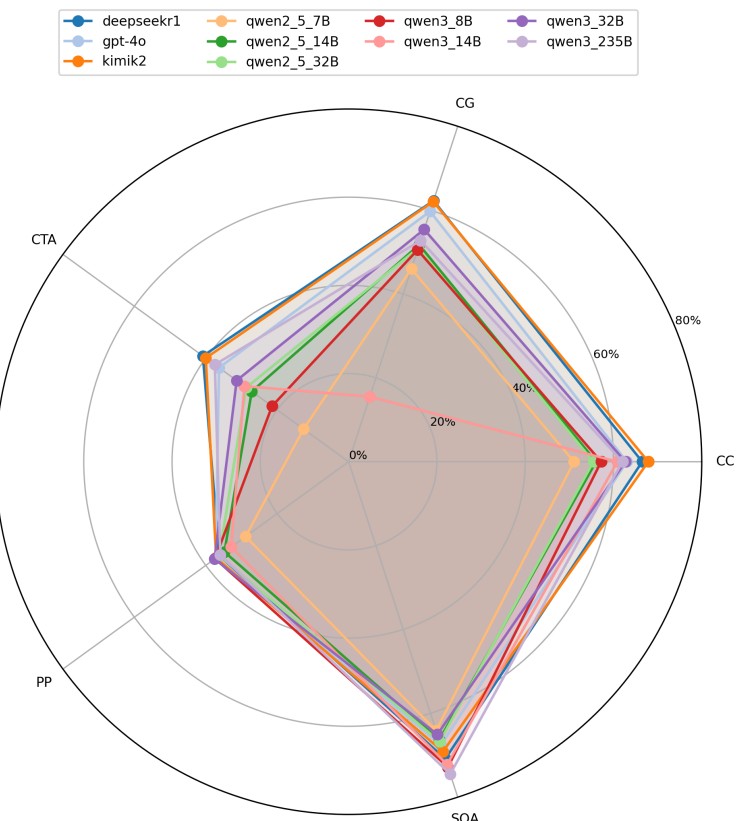

Figure 3: Radar-plot comparison of representative general-purpose models across the five SC-ARENA tasks: cell type annotation, perturbation prediction, cell generation, cell captioning, and scientific QA. The visualization highlights the uneven distribution of gains: while models such as Kimi-K2 and DeepSeek-R1 excel in captioning and generation, Qwen3-32B performs comparatively better in perturbation prediction. The radar plot provides a task-level perspective that complements aggregate scores and illustrates persistent challenges in mechanistic reasoning.

### A.8.2 DISCRIMINATIVE CAPACITY VIA ONTOLOGY PATH LENGTH TO ROOT.

To further validate the discriminative capacity of SC-ARENA, we examined the distribution of ontology path length to root for predicted cell types across models (Figure 4). Here, the x-axis represents binned intervals of the average path length to the ontology root, with shorter values corresponding to more specific and biologically precise annotations, while the y-axis reports the count of predictions falling into each interval.

The results reveal a clear scaling trend: Qwen3-32B produces deeper predictions on average, followed by Qwen3-14B and then Qwen3-8B. This hierarchy of prediction depth aligns closely with their overall benchmark performance, where larger models consistently outperform their smaller counterparts. Such consistency indicates that as models scale, they not only achieve higher aggregate scores but also tend to generate more specific and biologically meaningful predictions. This provides additional evidence that SC-ARENA can capture fine-grained distinctions in model behavior, delivering discriminative capacity beyond what traditional NLP metrics can offer.

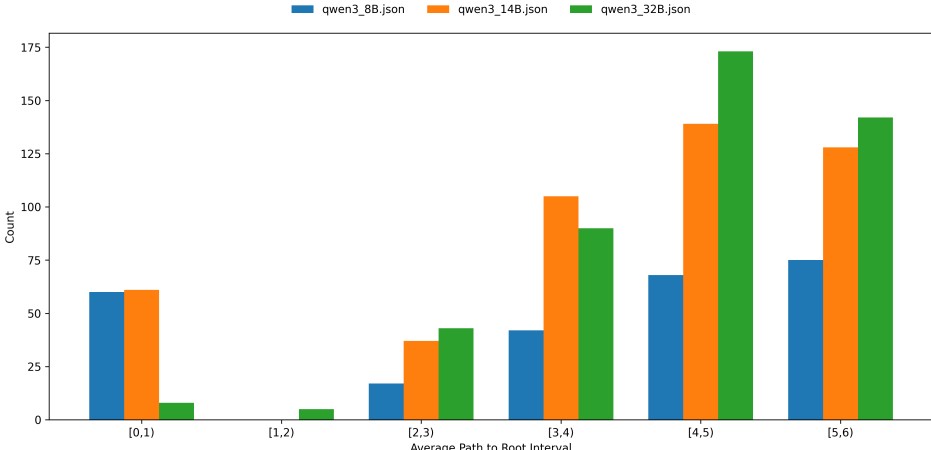

Figure 4: Distribution of ontology path length to root for predicted cell types across models. The x-axis shows binned intervals of the average path length to the ontology root, using left-closed, right-open notation [a,b), and the y-axis indicates the number of predicted cell types falling into each interval. Shorter path lengths indicate closer alignment with the ontology hierarchy and thus more specific predictions.

### A.8.3 ROBUSTNESS OF THE KNOWLEDGE-AUGMENTED EVALUATOR

To further validate the reliability of our knowledge-augmented evaluator, we conduct three additional robustness checks.

**(1) Cross-model evaluator consistency.**
We re-ran the full SC-ARENA benchmark using a second, independently trained judge model (`DeepSeek-V3.2-Exp`) as the evaluator, replacing the default `GPT-4o-mini`. The two evaluators demonstrate strong agreement across all five tasks and all tested models (metrics in Table 11). These results indicate that our evaluation outcomes are not artifacts of a specific LLM judge, but reflect stable, task-grounded judgments anchored in external biological knowledge.

**(2) Insensitivity to response length.**
We explicitly tested whether scoring is biased by answer verbosity. Across all model responses in SC-ARENA, we computed the correlation between token count and evaluation scores. The correlation is negligible (Table 11), confirming that our evaluator assesses content based on biological fidelity and reasoning quality, not surface form such as answer length.

**(3) Robustness to evaluation stochasticity.**
To assess stability against inherent randomness in the evaluation process, we re-ran the evaluation on the Cell Type Annotation (CTA) task with a different random seed (changing from `20250701` to `42`). The resulting scores remained highly consistent (Table 11), confirming that our evaluator's judgments are stable under typical stochastic variations encountered during LLM-based assessment.

Table 11: Quantitative metrics for robustness validation of the knowledge-augmented evaluator.

| Robustness Test | Spearman $\rho$ | Kendall's $\tau$ | Cosine Similarity |
| --- | --- | --- | --- |
| Cross-model evaluator consistency | 0.6701 | 0.5719 | 0.8873 |
| Insensitivity to response length | 0.0730 | 0.0578 | – |
| Robustness to evaluation stochasticity (CTA task) | 0.9070 | 0.8354 | 0.9468 |

### A.9 STEP-BY-STEP EXAMPLE OF KNOWLEDGE-AUGMENTED BIOLOGICAL PLAUSIBILITY SCORING

To enhance interpretability and clarify how biological plausibility is quantified in SC-ARENA, we present a concrete, end-to-end example from the Cell Type Annotation (CTA) task, corresponding to the evaluation logic illustrated in Figure 2b.

**Setup.** Consider the following instance:

- **Ground-truth cell type:** *CD16-positive, CD56-dim natural killer cell, human*
- **Model prediction:** *Natural Killer (NK) Cell*

**Step 1: Ontology Matching via OLS.** We standardize both labels using the Ontology Lookup Service (OLS) to map them to canonical identifiers in the Cell Ontology (CL) (Diehl et al., 2016):

- Ground-truth → `CL:2000001` (a terminal, highly specific NK cell subtype)
- Prediction → `CL:0000623` (the generic "natural killer cell" parent node)

**Step 2: Hierarchical Distance Computation.** We compute the shortest path length between the two CL nodes in the ontology graph. In this case:

$$\text{Distance} = 2$$

indicating that the predicted term is two levels above the ground-truth in the CL hierarchy—correctly capturing the NK lineage but lacking terminal granularity.

**Step 3: Mapping Distance to Plausibility Score.** While the raw distance itself is not the final score, it serves as a biologically grounded proxy for correctness. Our knowledge-augmented evaluator (an Eval-RAG-style LLM judge) uses this ontology path—along with functional definitions and marker gene context—to assign a human-interpretable rating on a 0–5 scale. In this example, the evaluator would typically assign a score of **3** (see Figure 10 for the full rubric), reflecting:

- Correct lineage identification (NK cell),
- Lack of terminal specificity (missing CD16+/CD56-dim distinction),
- Biological plausibility despite incompleteness.

**Step 4: Validation via Correlation Analysis.** Across all CTA predictions, we validate that such ontology-informed judgments align with human intuition. As shown in Figure 2a, evaluator scores exhibit a strong positive Spearman correlation with negative ontology distance:

$$\rho = 0.6212, \quad p < 0.001$$

This confirms that SC-ARENA's scoring is not arbitrary but systematically reflects biological relatedness encoded in curated ontologies.

This example illustrates how SC-ARENA transcends brittle string matching by integrating structured domain knowledge into every evaluation step—yielding scores that are both *interpretable* and *biologically faithful*.

### A.10 DETAILED PROMPT FOR EACH TASK

To ensure reproducibility and fairness, we provide here the full set of task-specific prompt templates used in SC-ARENA. For each benchmark task, we design two categories of prompts: (i) **answer generation prompts**, which are provided to the tested models to elicit predictions in a standardized format, and (ii) **score generation prompts**, which are presented to the evaluator model (GPT-4o-mini) to assign task-specific scores following the rubric in Appendix A.5. Together, these templates operationalize the Virtual Cell abstraction by unifying inputs, outputs, and evaluation across tasks.

**Cell Type Annotation (CTA).** The answer generation prompt instructs the model to infer the most likely ontology-grounded cell type from a ranked gene expression list (cell sentence). The evaluation prompt guides the judge to compare the predicted type with the gold label, rewarding exact matches or semantically close ontology categories.

**Cell Captioning (CC).** The answer generation prompt asks the model to produce a concise natural language description of the cell, highlighting marker genes and lineage context. The evaluation prompt checks whether the caption aligns with the ontology definition and expression evidence, penalizing vague or generic responses.

**Cell Generation (CG).** The answer generation prompt requires the model to synthesize a plausible ranked gene list given a cell type name and description. The evaluation prompt instructs the judge to verify consistency with marker gene databases and ontology knowledge, assigning partial credit when the generated profile is approximately correct.

**Perturbation Prediction (PP).** The answer generation prompt asks the model to predict both the perturbed cell sentence and sets of up- and down-regulated genes given a baseline cell sentence and perturbation condition. The evaluation prompt guides the judge to assess predictions against experimental ground truth and external references (e.g., NCBI, UniProt, GO), with scores reflecting both plausibility and mechanistic validity.

**Scientific QA (SQA).** The answer generation prompt presents the model with a domain-specific research question and asks for a step-by-step reasoning process leading to a concise final answer. The evaluation prompt provides the judge with the model's answer, the gold reference, and supporting PubMed context, instructing it to score factual accuracy, reasoning quality, and alignment with evidence.

Figures 5–9 illustrate the answer generation prompts for the five tasks, while Figures 10–14 present the corresponding evaluation prompts.

---

**Cell Type Annotation Answer Generation Prompt**

You are given {num_genes} genes ranked by expression level from a {organism} cell.

Cell sentence: {cell_sentence}Please reason step-by-step to determine the most probable cell type.

Consider known marker genes, expression patterns, and biological context.

After your reasoning, conclude with your prediction in the exact format: [Predicted_Cell_Type: ...]

---

Figure 5: Cell Type Annotation Answer Generation Prompt.

**Cell Captioning Answer Generation Prompt**

The following cell sentence represents {num_genes} genes from a {organism} cell, ranked by expression level.
Cell sentence: {cell_sentence}

Generate a concise, natural-language description of this cell that reflects the most specific cell type supported by the gene expression profile, while remaining biologically accurate and consistent with the Cell Ontology.

Guidelines:
- Prioritize lineage-defining markers, key functional modules, and unique biological roles directly indicated by the gene list.
- If gene evidence strongly supports a unique terminal cell type, describe it clearly and specifically.
- If evidence is insufficient or ambiguous for the exact terminal type, describe the most specific broader parent type supported by the data, and note the uncertainty or possible alternatives.
- Avoid generic phrases such as "highly active" or "robust metabolism" unless tied to specific markers.

Your entire response must be wrapped in the format: [Captioning: ...]

Figure 6: Cell Captioning Answer Generation Prompt.

**Cell Generation Answer Generation Prompt**

You are given the name and description of a cell type.
Cell Type: {cell_type}
Description: {cell_description}

Generate a cell sentence that reflects the expected gene expression profile of this cell. The cell sentence should be a comma-separated list of 200 genes, ordered from highest to lowest expression.

Ensure the gene ordering is biologically plausible and consistent with the described cell type's known functions and markers.

Be sure to state your answer using the exact format:  [Cell_Sentence: ...]

Figure 7: Cell Generation Answer Generation Prompt.

Perturbation Prediction Answer Generation Prompt

You are a single-cell transcriptomics expert.

Background
- Original expression profile (genes ranked by descending expression): {cell_sentence}
- Perturbation applied: {perturbation_description}Candidate differentially expressed genes (DEGs): {candidate_deg_list}

Task
1. From the provided candidate DEGs, identify which genes are significantly Up-regulated (Up) and which are Down-regulated (Down), based on the perturbation context and prior gene knowledge.
2. Based on these changes, generate the updated cell sentence that reflects the perturbed expression profile.

Output
- You may briefly explain your reasoning (≤ 5 bullets or ≤ 120 words).
- Conclude with the final answer in exactly this format, on a single line: [Up: geneA, geneB, ...][Down: geneX, geneY, ...][Cell_Sentence: gene1 gene2 gene3 ...]

Figure 8: Perturbation Prediction Answer Generation Prompt.

Scientific QA Answer Generation Prompt

You are a domain expert in single-cell biology. You will be given a specific type of question and the question itself. Please think step by step using relevant biological knowledge before answering. Your final answer must be enclosed in the following format: `[Answer: ...]` Use clear, concise, and scientifically accurate language.

Input:
Question Type: {question_type}
Question: {question}

Your Response:

Figure 9: Scientific QA Answer Generation Prompt.

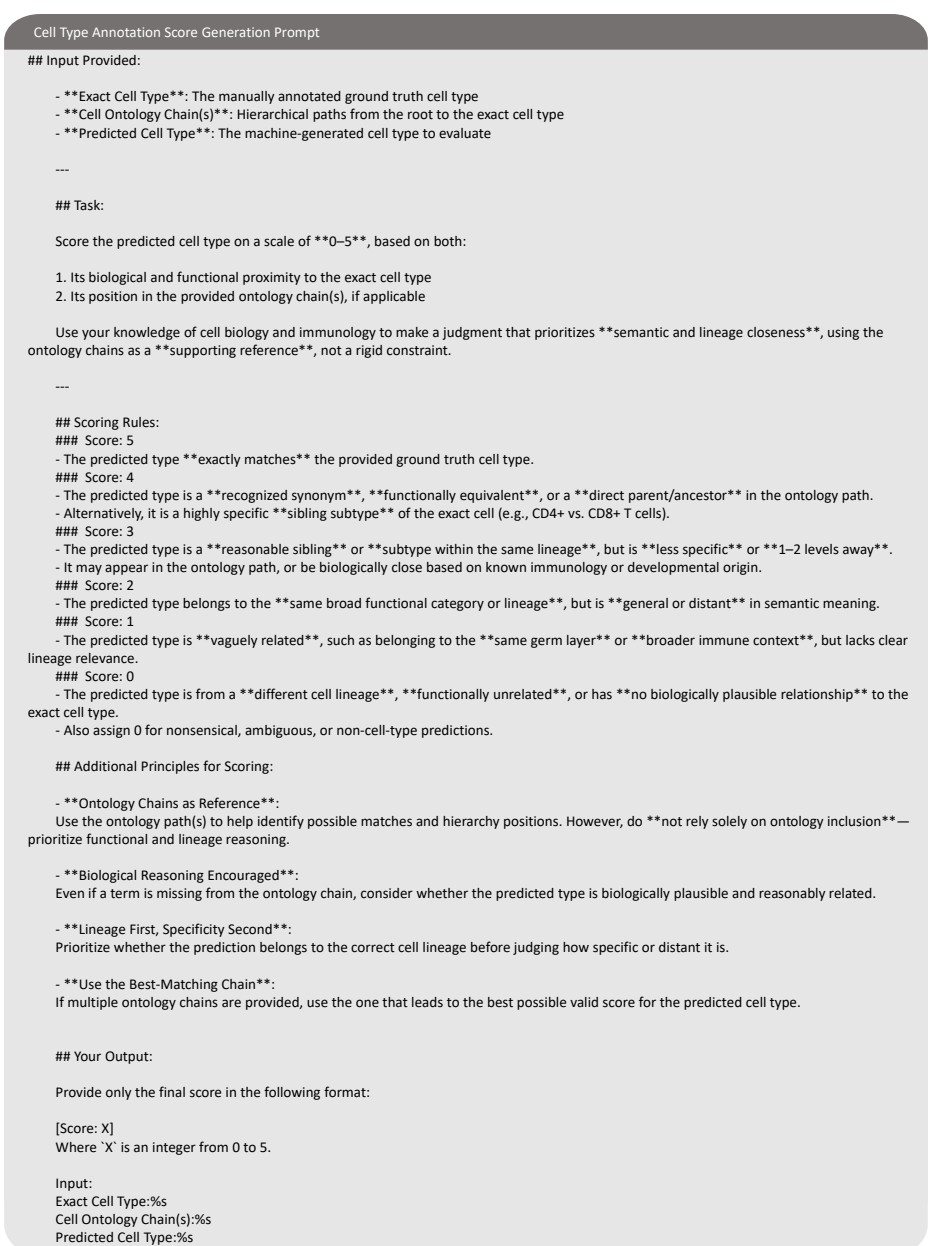

**Cell Type Annotation Score Generation Prompt**

## Input Provided:

- **Exact Cell Type**: The manually annotated ground truth cell type
- **Cell Ontology Chain(s)**: Hierarchical paths from the root to the exact cell type
- **Predicted Cell Type**: The machine-generated cell type to evaluate

---

## Task:

Score the predicted cell type on a scale of **0–5**, based on both:

1. Its biological and functional proximity to the exact cell type
2. Its position in the provided ontology chain(s), if applicable

Use your knowledge of cell biology and immunology to make a judgment that prioritizes **semantic and lineage closeness**, using the ontology chains as a **supporting reference**, not a rigid constraint.

---

## Scoring Rules:
### Score: 5
- The predicted type **exactly matches** the provided ground truth cell type.
### Score: 4
- The predicted type is a **recognized synonym**, **functionally equivalent**, or a **direct parent/ancestor** in the ontology path.
- Alternatively, it is a highly specific **sibling subtype** of the exact cell (e.g., CD4+ vs. CD8+ T cells).
### Score: 3
- The predicted type is a **reasonable sibling** or **subtype within the same lineage**, but is **less specific** or **1–2 levels away**.
- It may appear in the ontology path, or be biologically close based on known immunology or developmental origin.
### Score: 2
- The predicted type belongs to the **same broad functional category or lineage**, but is **general or distant** in semantic meaning.
### Score: 1
- The predicted type is **vaguely related**, such as belonging to the **same germ layer** or **broader immune context**, but lacks clear lineage relevance.
### Score: 0
- The predicted type is from a **different cell lineage**, **functionally unrelated**, or has **no biologically plausible relationship** to the exact cell type.
- Also assign 0 for nonsensical, ambiguous, or non-cell-type predictions.

## Additional Principles for Scoring:

- **Ontology Chains as Reference**:
Use the ontology path(s) to help identify possible matches and hierarchy positions. However, do **not rely solely on ontology inclusion**—prioritize functional and lineage reasoning.

- **Biological Reasoning Encouraged**:
Even if a term is missing from the ontology chain, consider whether the predicted type is biologically plausible and reasonably related.

- **Lineage First, Specificity Second**:
Prioritize whether the prediction belongs to the correct cell lineage before judging how specific or distant it is.

- **Use the Best-Matching Chain**:
If multiple ontology chains are provided, use the one that leads to the best possible valid score for the predicted cell type.

## Your Output:

Provide only the final score in the following format:

[Score: X]
Where `X` is an integer from 0 to 5.

Input:
Exact Cell Type:%s
Cell Ontology Chain(s):%s
Predicted Cell Type:%s

Figure 10: Cell Type Annotation Score Generation Prompt.

**Cell Captioning Score Generation Prompt**

You are a biomedical expert in single-cell transcriptomics and cell type classification, with deep expertise in the Cell Ontology and its hierarchical structure.

Your task is to **evaluate a model-generated description of a single cell** using three clearly separated inputs:

---

### 1. **Gene expression profile**
A ranked list of genes from most to least expressed.

{cell_sentence}

---

### 2. **Cell Ontology definition path**
A hierarchical lineage from a broad parent concept down to a specific, fine-grained cell type. Each level contains a name and definition.

{cell_path_chain}

---

### 3. **Cell description (to be evaluated)**
**IMPORTANT:** This is the only text produced by the model that you should score.
If this section is empty, contains only whitespace, or does not describe a cell type, you must assign **[Score: 0]** without further analysis.

{predicted_caption}

---

## **Evaluation Objective**
Assess whether the description in **Section 3** accurately and specifically reflects the **target cell type** as situated in the ontology path, while considering **gene expression evidence** from Section 1.

You must:
- Check **ontology match** (terminal node or appropriate ancestor).
- Check **gene expression support** for claimed specificity.

---

## **Key Principles**
- **5 points** — Description exactly matches the **terminal node**, supported by clear marker genes.
- **4 points** — Matches terminal node but with minor omissions; gene evidence mostly supportive.
- **3 points** — Correct broader parent type or plausible sibling type, supported by gene data; **does not** name terminal node.
- **2 points** — Overly broad or vague description with limited evidence.
- **1 point** — Barely relevant or generic tissue/system reference.
- **0 points** — Empty, unrelated, incoherent, or wrong cell type.

**Special rule:** If Section 3 is empty or generic (e.g., "unknown cell" / "this is a cell"), assign **0** immediately.

---

**Your answer should include a score in the following format:**
[Score: X]
Then add a brief justification (2–4 sentences) explaining the reasoning behind your score.

---

Figure 11: Cell Captioning Score Generation Prompt.

**Cell Type Annotation Score Generation Prompt**

## Input Provided:

- **Exact Cell Type**: The manually annotated ground truth cell type
- **Cell Ontology Chain(s)**: Hierarchical paths from the root to the exact cell type
- **Predicted Cell Type**: The machine-generated cell type to evaluate

---

## Task:

Score the predicted cell type on a scale of **0–5**, based on both:

1. Its biological and functional proximity to the exact cell type
2. Its position in the provided ontology chain(s), if applicable

Use your knowledge of cell biology and immunology to make a judgment that prioritizes **semantic and lineage closeness**, using the ontology chains as a **supporting reference**, not a rigid constraint.

---

## Scoring Rules:
### Score: 5
- The predicted type **exactly matches** the provided ground truth cell type.
### Score: 4
- The predicted type is a **recognized synonym**, **functionally equivalent**, or a **direct parent/ancestor** in the ontology path.
- Alternatively, it is a highly specific **sibling subtype** of the exact cell (e.g., CD4+ vs. CD8+ T cells).
### Score: 3
- The predicted type is a **reasonable sibling** or **subtype within the same lineage**, but is **less specific** or **1–2 levels away**.
- It may appear in the ontology path, or be biologically close based on known immunology or developmental origin.
### Score: 2
- The predicted type belongs to the **same broad functional category or lineage**, but is **general or distant** in semantic meaning.
### Score: 1
- The predicted type is **vaguely related**, such as belonging to the **same germ layer** or **broader immune context**, but lacks clear lineage relevance.
### Score: 0
- The predicted type is from a **different cell lineage**, **functionally unrelated**, or has **no biologically plausible relationship** to the exact cell type.
- Also assign 0 for nonsensical, ambiguous, or non-cell-type predictions.

## Additional Principles for Scoring:

- **Ontology Chains as Reference**:
Use the ontology path(s) to help identify possible matches and hierarchy positions. However, do **not rely solely on ontology inclusion**— prioritize functional and lineage reasoning.

- **Biological Reasoning Encouraged**:
Even if a term is missing from the ontology chain, consider whether the predicted type is biologically plausible and reasonably related.

- **Lineage First, Specificity Second**:
Prioritize whether the prediction belongs to the correct cell lineage before judging how specific or distant it is.

- **Use the Best-Matching Chain**:
If multiple ontology chains are provided, use the one that leads to the best possible valid score for the predicted cell type.

## Your Output:

Provide only the final score in the following format:

[Score: X]
Where `X` is an integer from 0 to 5.

Input:
Exact Cell Type:%s
Cell Ontology Chain(s):%s
Predicted Cell Type:%s

Figure 12: Cell Generation Score Generation Prompt.

**Perturbation Prediction Score Generation Prompt**

You are an expert in single-cell transcriptomics and gene regulation. Your task is to evaluate the **biological plausibility and accuracy** of a predicted gene expression perturbation in response to a specific condition. You will be given the following information:

1. **Unperturbed Cell Expression (cell sentence)** – A description of the gene expression profile before any perturbation.
2. **Perturbation Condition** – The experimental factor applied to perturb the cell.
3. **Ground Truth**:
- Perturbed Cell Expression (cell sentence)
- List of significantly **up-regulated genes**
- List of significantly **down-regulated genes**
4. **Predicted Result** by a language model:
- Predicted perturbed Cell Expression (cell sentence)
- Predicted significantly **up-regulated genes**
- Predicted significantly **down-regulated genes**
5. **Reference Knowledge**:
- Brief summaries from NCBI, GeneCards, and UniProt for the involved genes
- Gene Ontology (GO) information:
   - Cellular Component (GO_C_description)
   - Biological Process (GO_P_description)
   - Molecular Function (GO_F_description)

Your job is to compare the predicted response against the ground truth and reference knowledge, considering the following criteria:

- Are the **predicted expression changes** consistent with the true perturbation pattern?
- Are the **up/down-regulated genes** biologically plausible given the perturbation and consistent with known gene functions?
- Does the **predicted cell sentence** qualitatively resemble the real one in terms of key gene activity shifts?
- Are the predictions **supported or contradicted** by the provided reference knowledge?

Give a score from **0 to 5** based on overall plausibility and alignment with both ground truth and known biology, where:
- 0 = Completely incorrect and biologically implausible
- 1 = Poor prediction and unconvincing
- 2 = Somewhat plausible but with major gaps or errors
- 3 = Reasonable prediction with some soundness
- 4 = Mostly correct with minor inconsistencies
- 5 = Highly accurate and biologically consistent

**Your answer should include a score in the following format**:
`[Score: X]`

### Input:

1. **Unperturbed Cell Expression (cell sentence)**:
{ctrl_sentence}
2. **Perturbation Condition**:
{perturbation_description}
3. **Ground Truth**:
- **Perturbed Cell Expression (cell sentence)**:
{pert_sentence}
- **Up-regulated Genes**:
{up_genes_str}
- **Down-regulated Genes**:
{down_genes_str}
4. **Predicted Result** by the language model:
- **Predicted Perturbed Cell Expression (cell sentence)**:
{cell_sentence}
- **Predicted Up-regulated Genes**:
{up_genes}
- **Predicted Down-regulated Genes**:
{down_genes}
5. **Reference Knowledge**:
- **Gene Summaries (NCBI / GeneCards / UniProt)**:
{NCBI_gene_card_UniProt_summaries}
- **Gene Ontology Descriptions**:
- **Cellular Component**: {GO_C_description}
- **Biological Process**: {GO_P_description}
- **Molecular Function**: {GO_F_description}

Figure 13: Perturbation Prediction Score Generation Prompt.

---

**Scientific QA Score Generation Prompt**

You are a domain expert in single-cell biology and scientific reasoning.
Your task is to evaluate whether the answer provided by a language model ("Evaluated Model") to a scientific question is accurate, well-reasoned, and biologically sound.

## You will be given:
- **[Question Type]**: A label indicating the type of knowledge or reasoning required (e.g., Marker-Based Reasoning, Pathway Logic, Experimental Design, etc.).
- **[Original Question]**: The actual question that was posed to the model.
- **[Ground Truth Answer]**: A reliable, expert-verified reference answer.
- **[Model Answer]**: The output from the Evaluated Model.
- **[Reference Paper Title]**: The title of the scientific paper from which the question is derived.
- **[Reference Paper Abstract]**: The abstract of that paper, provided as external knowledge to help you assess correctness.
- **[Relevant Passage]**: The specific section of the paper most closely related to this question (may include results, figures, or methods). Use this passage as the primary reference for correctness.

## Instructions
Carefully analyze whether the Model Answer is:
1. **Scientifically correct** (check facts, terminology, biological mechanisms).
2. **Logically consistent** with the Original Question.
3. **Well-aligned** with the Ground Truth Answer.
4. **Appropriate** to the Question Type, showing the right reasoning depth and domain relevance.
5. **Consistent with and supported by the Reference Paper and Relevant Passage** (do not copy text verbatim, but use them to check correctness).

## Important Evaluation Rules
- Any **factual or scientific error** (e.g., misclassifying cytokines, incorrect pathway direction, or wrong biological effect) must lower the score.
- If such an error exists, the score **cannot be 5**.
- **Conceptual or mechanistic errors** that undermine reasoning (e.g., mixing up immune stimulatory vs suppressive roles) should be considered major flaws, scored **≤3**.
- If the answer is largely correct but contains **minor imprecision** (e.g., vague wording, lack of detail without scientific contradiction), it may be scored **4**.
- Only if the answer is **fully correct, with no scientific errors and strong alignment**, may it receive a **5**.

## Your Evaluation Should Include:

- **Strengths**: What the Model Answer did well.
- **Weaknesses / Errors**: Be explicit about what is wrong or misleading.
- **Impact of Errors**: How they affect correctness and scoring.

## Scoring Rubric
| Score | Description |
|-------|-------------|
| **5** | Fully correct, scientifically accurate, no errors, insightful, and well-aligned with the ground truth. |
| **4** | Mostly correct, but with minor flaws or imprecisions; no major scientific errors. |
| **3** | Partially correct, contains at least one clear scientific error or noticeable gap, though some correct reasoning is present. |
| **2** | Largely incorrect or incomplete; multiple scientific errors or major misunderstanding. |
| **1** | Minimally relevant, deeply flawed, or mostly wrong. |
| **0** | Completely incorrect, irrelevant, or nonsensical. |
At the end of your response, you must include the final score in this exact format:
`[Score: X]`

## Input
1. **Question Type**:
`{question["type"]}`
2. **Original Question**:
`{question["question"]}`
3. **Ground Truth Answer**:
`{question["answer"]}`
4. **Model Answer**:
`{model_answer}`
5. **Reference Paper Title**:
`{paper_title}`
6. **Reference Paper Abstract**:
`{paper_abstract}`
7. **Relevant Passage**:
`{question["relevant_passage"]}`

Figure 14: Scientific QA Score Generation Prompt.

