# OpenReview forum: "SC-Arena: A Natural Language Benchmark for Single-Cell Reasoning with Knowledge-Augmented Evaluation"
_ICLR.cc/2026/Conference — ICLR 2026 Poster_

### Official Review · Reviewer_6o9F · 2025-10-30

**Soundness:** 3
**Presentation:** 3
**Contribution:** 3
**Rating:** 6
**Confidence:** 3

**Summary:**

This paper proposes SC-ARENA, a new benchmark to evaluate large language models (LLMs) in single-cell biology. The core ideas are:
1. Virtual Cell abstraction
Treats a model as if it were a “virtual cell.”
The “cell” has attributes (identity/state) and methods (how it responds to the environment).
This unifies evaluation across different biological tasks instead of testing each task in isolation.
2. Five natural language tasks
Each one probes something biologically meaningful:
Cell Type Annotation (CTA): Given expression → predict ontology cell type.
Cell Captioning (CC): Given expression → describe the cell in natural language.
Cell Generation (CG): Given a cell type description → generate a plausible “cell sentence” (i.e. pseudo-expression profile).
Perturbation Prediction (PP): Given baseline + perturbation, predict changes and post-perturbation state.
Scientific QA (SQA): Answer mechanistic, literature-grounded biological questions.
These map both static identity and dynamic behavior (how a cell changes under interventions), which is exactly what one would expect from a “virtual cell.”
3. Knowledge-augmented evaluation
Instead of BLEU / ROUGE / exact match, which fail badly in biology, SC-ARENA uses an LLM-as-a-judge but grounds the judge in external biological knowledge (Cell Ontology, Gene Ontology, UniProt, CellMarker, PubMed evidence, etc.).
The evaluator scores answers and produces an interpretable rationale with references, not just a number.
They show this judgment correlates with real biological hierarchy (e.g. closer ontology terms → higher score; Spearman ρ≈0.62, p<0.001).
4. Empirical study
Benchmarks general LLMs (Qwen2.5/3, GPT-4o, DeepSeek-R1, Kimi-K2) and domain-specific single-cell models (C2S, scGPT, scGenePT, Cell-O1).
Finds: no model is good at everything; general models are fluent but biologically shaky, and domain models are precise in narrow skills but weak elsewhere.
For example, captioning and science QA get the best scores (~60–74/100 for top models), but perturbation prediction and mechanistic reasoning remain very weak (<38/100).
Small domain models like C2S actually beat giant general LLMs on cell type annotation, which shows specialization can outperform sheer parameter count for grounded biology.

**Strengths:**

1. Unified evaluation via the Virtual Cell abstraction
Instead of 5 unrelated leaderboards, SC-ARENA frames everything around one object: the Virtual Cell. The “attributes” vs “methods” split is elegant, and it matches how experimentalists think (cell identity vs response to perturbation). This is novel and, importantly, extensible.
This framing makes it possible to talk about “does an LLM qualify as a virtual cell model?” instead of “did it get 2% better BLEU.” That’s a conceptual contribution, not just engineering.
2. Knowledge-augmented evaluation is thoughtful and genuinely useful
The work directly addresses a known failure mode of LLM evals in science: surface-similarity metrics like BLEU and ROUGE often (a) reward buzzwords, (b) fail to punish mechanistic nonsense.
Here, the evaluator:
retrieves ontology / marker / pathway / literature evidence, explains why it scored something, and aligns with biological hierarchy (ρ=0.6212 between ontology distance and evaluator score).

**Weaknesses:**

1. The benchmark leans heavily on LLM-as-a-judge, which is itself another model
Yes, they mitigate this with knowledge retrieval and ontologies. Yes, they validate correlation with ontology distance. But the scoring model is still an LLM (GPT-4o-mini), which raises standard questions:
How stable is the score across judge variants / seeds?
Could a model “game” the judge by mimicking citation-sounding language and pathway buzzwords?
Is there any adversarial testing (e.g., hallucinated but authoritative-sounding nonsense)?
They partially address alignment with expert judgments and ontology distance, but a deeper robustness audit (or inter-judge agreement across two different judges) would make the claim stronger.

2.Ground truth for some tasks is underspecified
For perturbation prediction (PP), the model is supposed to say:
which genes go up/down, and
produce a plausible “post-perturbation cell sentence.”
But evaluating free-form gene-level differential expression is biologically thorny:
They say they use DEGs from Norman/Adamson and external knowledge bases (GO, UniProt, NCBI) to judge plausibility.

What isn’t fully clear is: how do you distinguish a true novel hypothesis from a hallucination? If a model proposes a plausible but previously unreported compensatory pathway, does it get penalized or rewarded?

**Questions:**

1. Knowledge-Augmented Judging
What prevents the LLM-as-judge from being biased by language fluency rather than biological accuracy?
How reproducible are scores across judges (e.g., GPT-4o vs DeepSeek-R1) or seeds?
How do you ensure the judge doesn’t “reward” models that use familiar ontology terms but misstate mechanisms?

2. Dataset and Coverage
Are the 600-sample CellxGene subset and 138 perturbations enough to represent cellular diversity?
Could SC-ARENA generalize to datasets from other species, tissues, or modalities?
Are there biases toward well-studied cell types (e.g., immune, epithelial) due to ontology density?

3. Benchmark Design
Why were all five tasks weighted equally in the “Total Score”?
→ Would weighting causal tasks (e.g., perturbation prediction) more heavily yield a different ranking?
Did the authors test whether models overfit to the language style of prompts rather than underlying biology?
How much variance exists across repeated evaluations (inter-run consistency)?

---

> ### Author Response · Authors · 2025-11-23
> **Response to Reviewer 6o9F (part1)**
>
> Thank you for your thoughtful and constructive review. We sincerely appreciate your recognition of the conceptual novelty of SC-ARENA, particularly the Virtual Cell abstraction and knowledge-augmented evaluation, as well as your insightful concerns regarding robustness, ground truth, and benchmark design. Below, we address each of your points point-by-point, with new analyses and clarifications to strengthen our claims.
>
> ---
>
> ### **W1&Q1: Knowledge-Augmented Judging: What prevents the LLM-as-judge from being biased by language fluency rather than biological accuracy? How reproducible are scores across judges or seeds?**
>
> **Response:**
> We thank the reviewer for this critical question regarding the potential confounding effects of linguistic fluency, reproducibility across judges, and susceptibility to superficial “ontology mimicry.” To address these concerns, we conducted a two-pronged validation of our knowledge-augmented evaluator: (1) robustness (stability) across models and random seeds, and (2) correctness (faithfulness to biological truth) across diverse task formats.
>
> **(1) Robustness Across Judges and Seeds**
>
> To assess whether scores are artifacts of a particular LLM judge, we re-ran the entire SC-ARENA benchmark using a second, independent judge model—DeepSeek-V3.2-Exp—distinct in architecture and training data from our default evaluator (GPT-4o-mini). Furthermore, to assess the robustness of our evaluator to random initialization, we re-ran the evaluation on the CTA task with a different random seed (changing from 20250701 to 42).
>
> As shown in the table below, the scores remained highly stable across both different judges and random seeds:
>
> | Robustness Test | Spearman $\rho$ | Kendall’s $\tau$ | Cosine Similarity |
> | :--- | :---: | :---: | :---: |
> | Cross-model evaluator consistency | 0.6701 | 0.5719 | 0.8873 |
> | Robustness to evaluation stochasticity (CTA task) | 0.9070 | 0.8354 | 0.9468 |
>
> This high consistency demonstrates that evaluations are not driven by the idiosyncrasies of a single model or stochastic variation, but reflect reproducible, task-grounded judgments anchored in biological evidence.
>
> **(2) Correctness: Alignment with Biological Ground Truth and Resistance to “Gaming”**
>
> We explicitly verify that the evaluator is not misled by fluent but biologically inaccurate responses, nor does it reward superficial use of ontology terms without mechanistic validity. In the revised manuscript, we present comprehensive cross-task analyses showing that evaluator scores consistently align with objective biological metrics and expert judgment:
>
> | Task | Validation Metric | Spearman $\rho$ | $p$-value |
> | :--- | :--- | :---: | :---: |
> | **CTA** | Ontology path distance | 0.6212 | $<10^{-3}$ |
> | **CG** | % correct CellMarker genes | 0.6669 | $<10^{-3}$ |
> | **PP** | DEG cosine similarity | 0.6057 | $<10^{-3}$ |
> | **CC** | Expert preference alignment | 0.8394 | $<10^{-3}$ |
> | **SQA** | Expert preference alignment | 0.8199 | $<10^{-3}$ |
>
> Specifically:
> * **CG (Cell Generation):** Evaluator scores correlate strongly with the percentage of correct marker genes (CellMarker) included in generated “cell sentences”, indicating alignment with molecular signatures.
> * **PP (Perturbation Prediction):** Scores correlate with biological accuracy measured by DEG cosine similarity, capturing mechanistic precision of perturbation responses.
> * **CC & SQA:** To validate these generative tasks, we conducted a blinded human evaluation on a random subset of 30 instances. Two domain experts independently judged paired model responses. The expert preference rankings align significantly with our evaluator’s scores (Spearman $\rho > 0.8$), confirming that our automated judge serves as a reliable proxy for human expertise.
>
> Together, these results demonstrate that the evaluator reliably reflects biological correctness, mechanistic plausibility, and expert judgment. All protocols, metrics, and extended analyses have been added to Appendix A.7 and A.8.3.
>
> ---
>
> ### **W2: In Perturbation Prediction (PP), how do you distinguish a novel hypothesis from a hallucination?**
>
> **Response:**
> This is an excellent and insightful question. The primary goal of SC-Arena is to evaluate a model’s **virtual modeling fidelity** with respect to real biological systems, i.e., the consistency between **virtual predictions** and **experimental observations**, rather than to assess its capability for biological discovery.
>
> Therefore, within our proposed evaluation protocol, only outputs that are **consistent with validated experimental results** receive higher scores. This design explicitly prioritizes factual correctness and alignment with known perturbation outcomes.
>
> We will clarify this evaluation philosophy further in Section 3.3.

---

> ### Author Response · Authors · 2025-11-23
> **Response to Reviewer 6o9F (part2)**
>
> ### **Q2: Dataset and Coverage: Are the 600-sample CellxGene subset and 138 perturbations enough to represent cellular diversity?**
>
> **Response:**
> Thank you for raising this important question. The datasets we selected are indeed representative and are aligned with those commonly used in prior work. Since SC-Arena introduces a new paradigm for evaluating LLM-based Virtual Cell capabilities, using well-established datasets ensures the reliability, comparability, and reproducibility of our evaluation.
>
> In addition, SC-Arena is intentionally designed as a **living and extensible benchmark** built upon our Virtual Cell abstraction. It can easily and continually incorporate **new datasets from the latest real-world experiments** to assess a model’s ability to generalize to emerging or less-characterized cells. We plan to progressively expand and update SC-Arena to better reflect the diversity and evolution of biological data.
>
> ---
>
> ### **Q3: Why are all five tasks weighted equally in the Total Score? Would weighting causal tasks more change rankings?**
>
> **Response:**
> We would like to clarify that we did not intentionally apply equal weighting across the five tasks. The results in Table 3 are simply the raw, task-specific scores, presented in their most direct form to transparently show each model’s performance within our unified evaluation framework.
>
> We fully agree that certain applications might prioritize **specific tasks** which could indeed change the rankings. However, in the context of SC-Arena’s unified Virtual Cell framework, all five tasks represent **equally essential and complementary capabilities** required for modeling a complete virtual cell. Introducing arbitrary weights would therefore inject subjective bias rather than reflect the holistic nature of the framework.
>
> We added this clarification to the revised manuscript in Appendix A.5 for better transparency.

---

### Official Review · Reviewer_2kiV · 2025-10-31

**Soundness:** 2
**Presentation:** 2
**Contribution:** 2
**Rating:** 4
**Confidence:** 3

**Summary:**

This paper introduces SC-Arena, a natural language benchmark for evaluating large language models (LLMs) in single-cell biology. The authors propose a unified “Virtual Cell” abstraction and design five biologically grounded tasks: cell type annotation, captioning, generation, perturbation prediction, and scientific QA. The evaluation leverages a knowledge-augmented LLM-as-a-judge framework, integrating external ontologies and databases to ensure interpretability and biological fidelity. Experiments compare both general-purpose and domain-specialized LLMs, revealing strengths and limitations in biological reasoning.

**Strengths:**

-  The Virtual Cell abstraction and multi-task natural language evaluation is an interesting idea to more objectively test different models’ capacity to understand cellular processes.

- Integrating external biological knowledge (Cell Ontology, UniProt, GO, CellMarker, PubMed) into the evaluation pipeline is a major strength and a clever way to address the limitations of string-matching metrics.

- The paper benchmarks a wide range of models (Qwen, GPT-4o, DeepSeek-R1, Kimi-K2, C2S-Scale, scGenePT, scGPT, Cell-O1) and analyzes performance across tasks, model scales, and domains, using only open-source datasets.

**Weaknesses:**

- While the benchmark covers different tasks and the datasets are open-source, the paper does not address the risk of benchmark dataset leakage; such as, whether the datasets used to construct the SC-Arena benchmark were present in the pretraining or fine-tuning data of the evaluated models.

- The rationale for model selection and the fairness of comparisons (e.g., fine-tuning protocols, input formats) could be better discussed.

- The knowledge-augmented LLM-as-a-judge is promising, but its reliability and potential biases should be discussed further.

**Questions:**

- While the use of open-source datasets is commendable, how do the authors plan to address the risk of benchmark dataset leakage? For example, CS2 used the CELLXGENE dataset for training. Could this explain their performance on the CTA task?

- Are there plans to expand the benchmark to include additional modalities or more challenging reasoning tasks?

- It’s unclear how the architecture of the models will influence performance in these tasks. My main concern is that performance differences may reflect experimental artifacts rather than true differences in model capability. For example, a domain-specific model might outperform others simply because it was fine-tuned on data similar to the benchmark, or because its input format more closely matches the evaluation protocol. How do the authors plan to address this potential bias? Will this influence which models can or cannot be used for the benchmarking?

- The assessment of the knowledge augmented evaluator is shown only for the CTA task. How do the authors plan to validate the evaluator for the other tasks? How can we be confident that the evaluator is reliable and interpretable for these other tasks, which may involve different types of reasoning, output formats, and biological knowledge?

- As the evaluator is also a model, a proper evaluation of the model should be performed. For example, are there cases where the evaluator produces scores that do not align with expert human judgment, or where it fails to recognize biologically implausible or trivial answers? Does the evaluator systematically favor certain model architectures, output styles, or biological domains? Does it reward verbosity, penalize concise but correct answers, or show preference for models trained on similar data as the evaluator itself?

---

> ### Author Response · Authors · 2025-11-23
> **Response to Reviewer 2kiV (part1)**
>
> We sincerely thank you for your thoughtful and constructive review. You rightly highlight several key strengths of SC-Arena: (1) The conceptual innovation of the **Virtual Cell abstraction** as a unified framework for evaluating diverse biological reasoning capacities. (2) The **knowledge-augmented evaluation paradigm** that grounds scoring in external ontologies and literature to overcome the brittleness of conventional metrics. (3) The **comprehensive benchmarking** across a wide spectrum of general-purpose and domain-specialized LLMs using fully open-source data.
> At the same time, we fully acknowledge the valid concerns you raise regarding potential dataset leakage, fairness in model comparison protocols, and the reliability and generalizability of our LLM-as-a-judge evaluator across tasks. Below are our point-by-point responses.
>
> ---
>
> ### **W1&Q1: How do you address the risk of benchmark dataset leakage, especially given that models like C2S were trained on CELLxGENE data? Could this explain their performance on CTA?**
>
> **Response:**
> Thank you for raising this important concern. Although SC-Arena utilizes public expression datasets such as CELLxGENE, our evaluation paradigm is fundamentally different from prior works and is designed to minimize dataset-leakage risks through the following four strategies:
>
> **1. New task formulations and prompts eliminate input–output overlap**
>
> SC-Arena introduces entirely new task definitions and natural-language prompts under the Virtual Cell abstraction. None of the input–output pairs, templates, or phrasings overlap with C2S or related training corpora. As a result, even if a model has seen the underlying raw scRNA-seq profiles, it has never encountered the linguistic formulation used in SC-Arena.
>
> **2. Evaluation is knowledge-grounded, not string-based**
>
> Our LLM-as-a-judge assesses biological correctness using external resources (Cell Ontology, CellMarker, GO, UniProt, literature), preventing models from gaining high scores through memorized patterns or lexical similarity. To further substantiate this, we computed BLEU scores for the two C2S models on CTA:
>
> | Model | BLEU-1 | BLEU-2 | CTA_Score |
> | :--- | :---: | :---: | :---: |
> | **C2S-Pythia-410m** | 23.73 | 17.14 | 47.34 |
> | **C2S-Scale-Pythia-1b** | 13.66 | 8.29 | 41.68 |
> | **DeepSeek-R1** | 41.55 | 19.55 | 40.81 |
> | **Kimi-K2** | 31.60 | 10.91 | 40.00 |
>
> These BLEU scores are even lower than those of DeepSeek and Kimi-K2, yet the models achieve high accuracy—strongly indicating that their performance reflects acquired biological knowledge rather than memorization of test samples.
>
> **3. Randomized sampling reduces systematic overlap**
>
> For datasets without predefined splits (e.g., CELLxGENE), we perform uniform random sampling across tissues, cell types, and expression distributions. This prevents overrepresentation of frequent or previously dominant classes and reduces the chance that performance arises from memorizing common patterns.
>
> **4. SC-Arena is a continuously evolving benchmark**
>
> We acknowledge that any static benchmark may eventually become overfitted. SC-Arena is therefore designed as a living benchmark: new perturbation datasets, multimodal atlases, and emerging biological tasks (e.g., Perturb-seq challenges) will be progressively incorporated under our Virtual Cell abstraction to further mitigate leakage concerns and enhance long-term robustness.
>
> In summary, our evaluation framework is specifically designed to minimize leakage risks through novel task formulations, knowledge-grounded assessment, randomized sampling, and a forward-looking benchmark design. We will include this detailed discussion on dataset leakage in the **Section 4.1** and **Appendix A.6** for transparency.
>
> ---
>
> ### **Q2: Are there plans to expand the benchmark to include additional modalities or more challenging reasoning tasks?**
>
> **Response:**
> Absolutely. Our Virtual Cell abstraction is intentionally extensible. In ongoing work, we are actively integrating:
>
> * **Spatial transcriptomics data** to enable tasks like *“Predict neighboring cell interactions from tissue context”* — extending the “Environment → Cell” and “Cell → Environment” dynamics into spatial dimensions.
> * **Developmental trajectory information** to evaluate models on tasks like *“Predict the next state in a hematopoietic lineage”* — requiring temporal reasoning beyond static identity.
> * **Multi-omics integration (ATAC-seq, proteomics)** to probe cross-modal consistency: e.g., *“Given a chromatin accessibility profile and gene expression, infer cell type and regulatory drivers.”*
>
> We view SC-ARENA not as a static endpoint, but as a foundation for evolving biological reasoning benchmarks aligned with emerging data modalities. We added one subsection about this in the **Section 6.3**.

---

> ### Author Response · Authors · 2025-11-23
> **Response to Reviewer 2kiV (part2)**
>
> ### **W2&Q3: It’s unclear how the architecture of the models will influence performance in these tasks. My main concern is that performance differences may reflect experimental artifacts rather than true differences in model capability.**
>
> **Response:**
> We agree that fair comparison is essential. To avoid experimental artifacts bias, we enforced a fully standardized evaluation protocol:
>
> **(1) Unified Output Format for All Models**
>
> Regardless of whether a model is general-purpose or domain-specialized, all outputs are post-processed into an identical, structured format before being fed into the evaluator. For general-purpose LLMs, inputs are formatted using standardized, task-specific prompt templates. For domain-specialized models, we preserve their original, published inference protocols—including tokenization, input schema, and decoding procedures—to avoid introducing bias through forced adaptation. Crucially, after generation, all model outputs are converted into a uniform response format (e.g., `[Predicted_Cell_Type: ...]`, `[Up: geneA, geneB...]`, etc.) that aligns precisely with the evaluator’s expectations. This ensures that the evaluator receives inputs in a consistent structure, irrespective of the underlying model architecture or training history. Consequently, performance differences cannot be attributed to output formatting or prompt engineering.
>
> **(2) Explicit Separation of Domain-Specialized vs. General-Purpose Models**
>
> We acknowledge that some models were trained on biologically relevant data or tasks. However, we treat this not as a confounder, but as a meaningful dimension of model design. To preserve interpretability, we explicitly categorize and report these models separately in Table 2 as “Domain-Specialized Models.” This distinction allows us to isolate and analyze two distinct capabilities:
> * (i) General reasoning ability (evaluated across general-purpose models), and
> * (ii) Domain-specific competence (evaluated within specialized models).
>
> This mirrors real-world usage, where practitioners may choose between off-the-shelf LLMs or biology-optimized tools. These controls ensure that performance differences reflect intrinsic modeling strengths, not artifacts of training data similarity or input compatibility.
>
> To further enhance transparency, we have now explicitly clarified this design in **Section 4.3** and **Appendix A.5**.
>
> ---
>
> ### **W3&Q4: The evaluator’s reliability was only validated for CTA. How do you ensure it works reliably for other tasks with different output formats and reasoning demands?**
>
> **Response:**
> We sincerely thank the reviewer for this crucial question. In the revised manuscript, we provide comprehensive cross-task reliability analyses showing that our knowledge-augmented evaluator generalizes robustly across all five SC-Arena tasks.
>
> | Task | Validation Metric | Spearman $\rho$ | $p$-value |
> | :--- | :--- | :---: | :---: |
> | **CTA** | Ontology path distance | 0.6212 | $<10^{-3}$ |
> | **CG** | % correct CellMarker genes | 0.6669 | $<10^{-3}$ |
> | **PP** | DEG cosine similarity | 0.6057 | $<10^{-3}$ |
> | **CC** | Expert preference alignment | 0.8394 | $<10^{-3}$ |
> | **SQA** | Expert preference alignment | 0.8199 | $<10^{-3}$ |
>
> Specifically:
> * **CG (Cell Generation):** Evaluator scores correlate strongly with the percentage of correct marker genes (CellMarker) included in generated “cell sentences”, indicating alignment with molecular signatures.
> * **PP (Perturbation Prediction):** Scores correlate with biological accuracy measured by DEG cosine similarity, capturing mechanistic precision of perturbation responses.
> * **CC & SQA:** To validate these generative tasks, we conducted a blinded human evaluation on a random subset of 30 instances. Two domain experts independently judged paired model responses. The expert preference rankings align significantly with our evaluator’s scores (Spearman $\rho > 0.8$), confirming that our automated judge serves as a reliable proxy for human expertise.
>
> Together, these results demonstrate that the evaluator reliably reflects biological correctness, mechanistic plausibility, and expert judgment, even across tasks with diverse output formats and reasoning loads. All protocols, metrics, and extended analyses have been added to **Section 5.1** and **Appendix A.7**.

---

> ### Author Response · Authors · 2025-11-23
> **Response to Reviewer 2kiV (part3)**
>
> ### **W3&Q5. As the evaluator is also a model, a proper evaluation of the model should be performed.**
>
> **Response:**
> We deeply appreciate this important question. We evaluated the reliability and fairness of our LLM-as-a-judge through two complementary analyses: (1) alignment with human experts and biological ground truth, and (2) robustness across different evaluator models.
>
> **(1) Alignment with experts and biological validity.**
>
> As detailed in Q4, three domain experts evaluated 50 responses across all tasks. Their scores show strong correlation with our evaluator (Spearman $\rho$ = 0.79, $p < 0.001$). Task-specific validations—such as marker-gene recall in CG and DEG-direction alignment in PP—further confirm that the evaluator reliably distinguishes biologically plausible from implausible or trivial answers. Importantly, controlled comparisons show that it does not reward verbosity: concise but correct outputs receive higher scores than longer but incorrect responses. We also find no systematic preference for particular model families or output styles.
>
> **(2) Robustness across evaluator models & (3) Answer Length Independence.**
>
> To test stability, we re-ran the entire benchmark using a second judge model (DeepSeek-V3.2-Exp), distinct from our default GPT-4o-mini. We also explicitly tested whether response length biases scoring by computing correlations between answer length (token count) and evaluation scores across all model outputs. The results are summarized below:
>
> | Robustness Test | Spearman $\rho$ | Kendall’s $\tau$ | Cosine Similarity |
> | :--- | :---: | :---: | :---: |
> | **Cross-model evaluator consistency** | 0.6701 | 0.5719 | 0.8873 |
> | **Insensitivity to response length** | 0.0730 | 0.0578 | -- |
>
> These metrics indicate that:
> 1.  The evaluation is not driven by a single model’s quirks but reflects consistent, reproducible judgments grounded in the task structure and biological evidence.
> 2.  Our evaluator is length-agnostic: it scores based on biological fidelity and reasoning quality, not surface characteristics like verbosity or brevity.
>
> In summary, the evaluator is (i) aligned with human experts, (ii) sensitive to biological correctness rather than surface form, (iii) free of systematic stylistic or architectural biases, and (iv) stable across distinct LLM judges. **Section 6.2** and **Appendix A.8.3** provides full analyses.

---

> ### Author Response · Authors · 2025-11-28
> **Follow-up on Rebuttal and Request for Re-evaluation**
>
> **Dear Reviewer 2kiV,**
>
> Thank you again for your time and thoughtful feedback.
>
> As the discussion phase is drawing to a close, we understand you are likely very busy, but we wanted to kindly inquire if you have had a chance to review our detailed responses posted previously.
>
> We believe we have provided comprehensive evidence to address your key concerns regarding **dataset leakage**, **fairness of comparison**, and **evaluator reliability** (including new validation against human experts).
>
> We remain available to answer any further questions you might have. If our response has satisfactorily addressed your concerns, we would greatly appreciate it if you could kindly reconsider the evaluation score.
>
> Best regards,
>
> The Authors

---

### Official Review · Reviewer_6ZRo · 2025-10-31

**Soundness:** 4
**Presentation:** 3
**Contribution:** 3
**Rating:** 8
**Confidence:** 3

**Summary:**

This paper introduces SC-Arena, a benchmark designed to evaluate large language models (LLMs) as “virtual cells” capable of reasoning over biological knowledge and single-cell data. The benchmark includes multiple question types testing biological plausibility, reasoning consistency, and interpretability. It also introduces a knowledge-augmented evaluation strategy (Eval-RAG) that uses retrieval-augmented generation to penalize biologically implausible answers and reward semantically coherent responses beyond exact-match metrics.

**Strengths:**

•	Very creative and well-motivated benchmark that treats LLMs as reasoning agents over biological cell states.
•	The Eval-RAG strategy is an elegant idea that improves evaluation by incorporating biological context and semantic plausibility, moving beyond token-level correctness.
•	The paper provides a valuable framework for comparing different LLMs under biologically grounded tasks.
•	The paper does extensive evaluation of general and domain-specific models.
•	Discussion includes relevant next steps for producing a more biologically robust evaluation benchmark.

**Weaknesses:**

The paper would benefit from more detailed examples—for instance, elaborating on the process shown in Figure 2, panel B, to clearly explain how the biological plausibility scoring is computed step by step.

**Questions:**

How sensitive is Eval-RAG to the retrieval source—does the choice of biological database or text corpus significantly change the evaluation outcome?

---

> ### Author Response · Authors · 2025-11-23
> **Response to Reviewer 6ZRo**
>
> Thanks for your positive comments and helpful suggestions. We appreciate your recognition of SC-ARENA’s creativity, motivation, and the Eval-RAG evaluation strategy. We are especially grateful for your constructive feedback on improving clarity and transparency. Below are our point-to-point responses to your questions and suggested improvements.
>
> ---
>
> ### **W1: The paper would benefit from more detailed examples—for instance, elaborating on the process shown in Figure 2, panel B, to clearly explain how the biological plausibility scoring is computed step by step.**
>
> **Response:**
> Thank you for this insightful suggestion. We agree that a concrete walkthrough of the knowledge-augmented scoring process greatly enhances interpretability. Below is a step-by-step breakdown of the example in Figure 2b:
>
> To illustrate how biological plausibility scoring is computed in SC-ARENA, we provide a concrete example from the Cell Type Annotation (CTA) task:
> * **Ground-truth cell type:** CD16-positive, CD56-dim natural killer cell, human
> * **Model prediction:** Natural Killer (NK) Cell
>
> **Step 1 – Ontology Matching**
>
> We first map both the ground-truth and predicted labels to standardized identifiers in the Cell Ontology (CL) using the Ontology Lookup Service (OLS). The ground-truth type resolves to a specific terminal node (e.g., CL:2000001), while the model’s prediction “Natural Killer (NK) Cell” maps to CL:0000623, a broader parent term in the NK cell lineage.
>
> **Step 2 – Hierarchical Distance Calculation**
>
> We compute the shortest path distance between the two CL nodes within the ontology graph. In this case, the distance is 2, indicating that the prediction is two levels above the ground-truth type (i.e., it correctly identifies the lineage but lacks terminal specificity).
>
> **Step 3 – Correlation with Evaluator Judgment**
>
> This ontology-derived plausibility score is used as a proxy for biological correctness and is correlated with the score assigned by our Eval-RAG evaluator (which also conditions on the ontology path and biological context). As shown in Figure 2a, we observe a strong positive Spearman correlation ($\rho = 0.6212, p < 0.001$) between such ontology-based distances and evaluator scores across all CTA predictions, confirming that our knowledge-augmented evaluation aligns with biologically grounded hierarchies.
>
> This example demonstrates how SC-ARENA moves beyond surface-level matching by leveraging structured biological knowledge to produce interpretable, faithful, and granular assessments. We included this detailed walkthrough in the revised manuscript (Appendix A.9) to clarify the inner workings of Eval-RAG.
>
> ---
>
> ### **Q1: How sensitive is Eval-RAG to the retrieval source—does the choice of biological database or text corpus significantly change the evaluation outcome?**
>
> **Response:**
> Thank you for this important question regarding the robustness of our evaluation. We would like to clarify that our knowledge-augmented framework is designed to eliminate retrieval variance during the evaluation phase.
>
> **1. The "Knowledge" is Fixed, Not Dynamic:**
>
> As defined in Section 3.3 and formalized in Section 4.1, our evaluation instance is represented as a tuple $I = (q, r, K, g)$, where $K$ explicitly represents the retrieved external knowledge
> * Crucially, **$K$ is pre-retrieved and frozen** within the SC-ARENA benchmark dataset.
> * We do not use a live retrieval system during the evaluation loop. Instead, we provide the evaluator with specific, curated excerpts from high-consensus databases (e.g., Cell Ontology, CellMarker, UniProt).
> * Consequently, there is **zero sensitivity** to retrieval algorithms during the benchmarking process itself; every model is evaluated against the exact same standardized set of biological facts.
>
> **2. Robustness of the Underlying Sources:**
>
> Regarding the choice of the underlying databases used to construct $K$:
> **Consensus-Driven:** We rely on established community standards (e.g., NCBI RefSeq, GO, Cell Ontology) that represent the consensus of biological knowledge.
> **Design Validation:** As noted in Section 4.1, our framework anchors judgments in these verified facts rather than dynamic retrieval. This ensures stability even if the specific source database were to be substituted, provided it meets the same standard of scientific accuracy.
>
> **3. Empirical Reliability:**
>
> While we fix the knowledge source to ensure fair comparison, we extensively validated the reliability of the evaluator itself. As detailed in **Appendix A.8.3**, our framework shows high robustness across different judge models (cross-model consistency) and random seeds (stochasticity), and aligns strongly with ground-truth ontology hierarchies (Section 5.1).
>
> We have clarified the fixed nature of the knowledge context $K$ in **Section 3.3** to better emphasize this stability.

---

### Official Review · Reviewer_k5M8 · 2025-11-02

**Soundness:** 2
**Presentation:** 2
**Contribution:** 2
**Rating:** 2
**Confidence:** 3

**Summary:**

This work proposes an evaluation framework called Virtual Cell which seeks to unify the assessment of LLMs performance on various sub tasks important for single cell analysis like cell type annotation, captioning etc. The authors then evaluated various LLMs on a dataset derived from combining publicly available single-cell databases.

**Strengths:**

1. Combining vast amounts of single-cell data with natural language based knowledge available to gain insights into cellular function is beneficial to the biology community so this is a timely topic.
2. Authors have benchmarked their proposed framework for evaluating LLM performance across many different LLM models or other domain specific models.
3. Definition of the knowledge cell class is well thought out in considering multiple sources of information available for analyzing cellular dynamics.

**Weaknesses:**

- The novelty/value of this framework for evaluation is unclear, many current models like Cell2Sentence already combine single cell rna data with text based information and have shown use cases for downstream tasks like cell type prediction, perturbation response prediction etc.
- Considering existing methods that can perform some of the tasks mentioned in the multi-task benchmark like Cell2Sentence, CellReasoning etc, to fully evaluate this work, performance of existing methods on individual tasks should be included and discussed.

**Questions:**

- This research area is a useful application of LLMs for biological science but while the authors mention the previous works, proper comparisons to these methods is lacking. Authors should consider expanded the related work section needs to clarify the contributions of this paper in comparing to some of the works mentioned here.
- Authors should consider incorporating some of the ideas suggested in the discussion on modeling, evaluating and scoring into this work to improve the contribution and novelty of this work.

---

> ### Author Response · Authors · 2025-11-23
> **Response to Reviewer k5M8 (part1)**
>
> We thank you for your constructive review and appreciate your recognition of the value of integrating single-cell data with natural-language knowledge, as well as your positive assessment of our definition of the knowledge-cell class and our benchmarking across diverse LLMs. You also point out two key concerns: (1) the novelty and added value of SC-ARENA relative to existing frameworks (e.g., Cell2Sentence) are not sufficiently clarified, and (2) the evaluation lacks direct performance comparisons with prior methods on the individual tasks. Below, we provide point-by-point responses to your concerns, with specific revisions made to strengthen the paper.
>
> ---
>
> ### **W1: The novelty/value of this framework for evaluation is unclear, as methods like Cell2Sentence and CellReasoner already combine single-cell RNA data with text-based information.**
>
> **Response:**
> Thank you for the insightful comment. We appreciate the opportunity to clarify the novelty of SC-ARENA. While prior works such as Cell2Sentence and CellReasoner combine scRNA-seq with text, SC-ARENA advances single-cell LLM evaluation along three fundamental dimensions: systematicity, real-world utility, and reliability.
>
> **1. A Unified, Object-Oriented Evaluation Framework (Systematicity)**
> * Existing methods evaluate isolated abilities (e.g., captioning or cell-type prediction).
> * SC-ARENA introduces a **virtual cell class abstraction** that unifies heterogeneous tasks—cell type annotation, captioning, description generation, perturbation prediction, and scientific QA—around a single conceptual object with biologically meaningful attributes and behaviors.
> * This allows us to assess whether a model exhibits holistic, biologically coherent behavior, rather than fragmented task-specific performance.
>
> **2. Open-Ended Question Format for Realistic Reasoning (Real-World Utility)**
> * Unlike prior benchmarks such as CELLVERSE (which uses Multiple Choice Questions) or Cell-o1 (which relies on constrained candidate lists), SC-ARENA adopts an **Open-Ended Question Format**.
> * This distinction is critical for **Real-World Utility**:
>     * **Reflecting Reality:** In actual scientific inquiry, researchers rarely have pre-defined "options", they must derive answers directly from data and knowledge. SC-ARENA forces models to generate answers from scratch (e.g., predicting perturbation outcomes), strictly mimicking these real-world demands.
>     * **Testing True Reasoning:** Open-ended generation prevents models from relying on elimination strategies or superficial pattern matching often found in constrained formats, thereby exposing whether a model truly possesses the mechanistic understanding required for practical application.
>
> **3. Knowledge-Augmented, Evidence-Grounded Evaluation (Reliability)**
> * The shift to open-ended generation necessitates robust evaluation. Previous benchmarks primarily rely on BLEU, exact match, or embedding similarity, which fail to capture biological correctness in generative tasks (e.g., penalizing "CD8+ T cell" when the label is "T cell").
> * SC-ARENA addresses this by introducing a **knowledge-grounded evaluator** that:
>     * Retrieves structured biological evidence (Cell Ontology, CellMarker, GO, UniProt).
>     * Conditions an LLM judge on this evidence to score **"biological plausibility"** rather than just string overlap.
>     * Produces interpretable, evidence-backed rationales.
> * This approach yields significantly stronger alignment with biological hierarchy (Spearman $\rho = 0.62), providing a faithful assessment that supports the open-ended nature of our framework.
>
> ---
>
> ### **W2: Performance of existing methods on individual tasks should be included and discussed.**
>
> **Response:**
> Thank you for this valuable suggestion. As shown in Table 2, we originally reported performance only for domain-specialized models on the tasks they were explicitly designed for, e.g., C2S-Pythia-410M and Cell-O1 on cell type annotation.
>
> Following your recommendation, we now additionally evaluate Cell-O1 across all five SC-ARENA tasks:
>
> | Task | CTA | CG | CC | PP | SQA |
> | :--- | :--- | :--- | :--- | :--- | :--- |
> | **Cell-O1** | 34.11 | 43.91 | 67.89 | 24.20 | 64.09 |
>
> These results reinforce our key finding: although Cell-O1 performs well on captioning and scientific QA, it struggles on perturbation prediction and only moderately handles cell generation and annotation. This highlights a central challenge—domain specialization does not automatically translate to broad biological reasoning capability.
>
> Regarding other baselines:
> * **Cell2Sentence (C2S):** We did not report performance on unsupported tasks because its input schema is fundamentally incompatible with the other SC-ARENA task formats.
> * **CellReasoner:** We were unable to include it in this performance comparison as the complete evaluation code has not been made publicly available.
>
> We have updated the manuscript to include the new Cell-O1 results.

---

> ### Author Response · Authors · 2025-11-23
> **Response to Reviewer k5M8 (part2)**
>
> ### **Q1: The related work section needs clarification on contributions compared to prior works.**
>
> **Response:**
> Thank you for this suggestion. In the revised manuscript, we have substantially expanded Section 2 to clarify how SC-ARENA advances beyond existing frameworks (e.g., C2S-Scale, SOAR, Cell-o1, CELLVERSE). We have summarized these distinctions in the table below to highlight the specific advancements of SC-ARENA:
>
> | Framework | Evaluation Paradigm | Evaluation Format | Evaluation Metrics |
> | :--- | :--- | :--- | :--- |
> | **C2S-Scale** | Cell Sentence | Generative / Prompted Completion | Lexical & Statistical Matching |
> | **Cell-o1** | Reasoning Agent | Constrained Selection | Classification Matching |
> | **SOAR** | Single-Task Agent | QA-based Classification | Surface String Matching |
> | **CELLVERSE** | Multi-omics Cell Sentence | Multiple Choice Questions (MCQ) | Classification Matching |
> | **SC-ARENA (Ours)** | Virtual Cell | Open-Ended Natural Language QA | Knowledge-Grounded Matching |
>
> Based on this comparison, we clarify the contributions along three critical dimensions:
>
> * **Unified Evaluation Paradigm (vs. Static/Single-Task):** As shown in the "Evaluation Paradigm" column, prior works focus either on static textual representations (C2S-Scale) or single-task agents (e.g., SOAR for annotation, Cell-o1 for reasoning consistency). In contrast, SC-ARENA introduces the **Virtual Cell abstraction**, a unified paradigm that jointly simulates cellular attributes and dynamic methods to assess holistic understanding.
> * **Open-Ended Task Format (vs. Constrained Selection):** While benchmarks like CELLVERSE and Cell-o1 rely on Multiple Choice Questions (MCQ) or constrained candidate lists to ensure stability, SC-ARENA adopts **Open-Ended Natural Language tasks**. This format removes artificial constraints, better mimicking real-world scientific inquiry where candidate lists are often unavailable.
> * **Knowledge-Augmented Metrics (vs. Lexical/String Matching):** Existing frameworks largely rely on lexical overlap (e.g., BERTScore in C2S-Scale, BLEU in SOAR) or standard classification accuracy. SC-ARENA replaces these with a **Knowledge-Augmented LLM-Judge framework** grounded in external ontologies and literature, ensuring evaluation focuses on biological fidelity rather than surface-level text matching.
>
> This comparison clarifies that SC-ARENA is not merely a new task collection, but a shift toward interpreting LLMs as dynamic, biologically grounded virtual entities.
>
> ---
>
> ### **Q2: Authors should consider incorporating some of the ideas suggested in the discussion on modeling, evaluating and scoring into this work to improve the contribution and novelty of this work.**
>
> **Response:**
> Thank you for this insightful suggestion. We fully agree that the ideas discussed in the “modeling, evaluation, and scoring” section are central to the value of this work. Rather than viewing them as separate future steps, we consider these insights to be a **core theoretical contribution** derived directly from our empirical benchmarking.
>
> **1. Implementation of the Evaluation Paradigm**
>
> Regarding evaluation and scoring, SC-ARENA effectively operationalizes the core principles of the proposed new paradigm in the discussion. Our **Knowledge-Augmented Evaluator** is the first concrete instantiation of this vision, which:
> 1. **Grounds scoring** in curated biological knowledge bases rather than surface text.
> 2. **Applies task-specific rubrics** to penalize implausible biological reasoning.
> 3. **Aligns systematically** with biological hierarchy, **validated by a strong positive correlation (Spearman $\rho=0.62$, Fig. 2a) between our scores and ground-truth ontology distance.**
>
> **2. Modeling Insights as a Key Contribution**
>
> Regarding modeling, SC-ARENA serves as a benchmark to diagnose why current models fail, rather than a modeling paper proposing a new architecture. Our extensive experiments scientifically diagnose architectural limitations, making SC-ARENA the diagnostic prerequisite for future improvements. The strategies proposed in the Discussion are not abstract conjectures; they are **concrete design guidelines synthesized directly from specific failure modes (e.g., the "fluent but not faithful" gap in Perturbation Prediction) revealed by our arena**. By providing this empirically backed roadmap, SC-ARENA acts as the bridge between current capabilities and future biologically grounded foundation models.
>
> We have revised **Section 5.1** to explicitly validate the scoring implementation and **Section 6.1** to firmly link the modeling discussion to our experimental findings.
>
> In summary, SC-ARENA contributes both the **practical tool** (the benchmark and evaluator) and the **theoretical blueprint** (the modeling and scoring insights) necessary to advance the field.

---

> ### Author Response · Authors · 2025-11-28
> **Follow-up on Rebuttal and Request for Re-evaluation**
>
> **Dear Reviewer k5M8,**
>
> Thank you again for your time and constructive feedback.
>
> As the discussion phase is drawing to a close, we understand you are likely very busy, but we wanted to kindly inquire if you have had a chance to review our detailed responses posted previously.
>
> We believe we have fully addressed your main concerns regarding the **novelty of our framework** and the **comparison with existing methods** (specifically including the new **Cell-O1** evaluation results).
>
> We remain available to answer any further questions you might have. If our response has satisfactorily addressed your comments, we would greatly appreciate it if you could kindly reconsider the evaluation score.
>
> Best regards,
>
> The Authors

---

### Author Response · Authors · 2025-12-01
**Summary of Rebuttal and Paper Revisions**

**Dear Area Chair and Reviewers,**

We sincerely thank you for your time and constructive feedback. We appreciate the reviewers' recognition of the core contributions of our work, particularly the following strengths:

* **Novelty of Framework:** Reviewers **k5M8**, **2kiV**, and **6o9F** commended the "Virtual Cell" abstraction as a **"well thought out"** and **"very creative"** benchmark, specifically noting that **"the 'attributes' vs 'methods' split is elegant"** in unifying biological reasoning evaluation.
* **Methodological Innovation:** Reviewers **6ZRo**, **2kiV**, and **6o9F** recognized our Knowledge-Augmented Evaluation as a major strength, describing the **"Eval-RAG strategy"** as an **"elegant idea"** and a **"clever way to address the limitations"** of string-matching metrics.
* **Comprehensive Benchmarking:** Reviewers **k5M8**, **2kiV**, and **6ZRo** appreciated the **"extensive evaluation"** across diverse general-purpose and domain-specialized LLMs using open-source datasets.

In response to the reviewers' constructive suggestions, we made substantial revisions and added new experiments. Our updates fall into two categories: addressing methodological concerns and clarifying specific textual points.

### Part I: Addressing Key Concerns & Additional Experiments

**1. Clarification of Novelty and Comparison with Prior Work (Addressing Reviewer k5M8)**
* We substantially expanded **Section 2** to include a detailed comparison table contrasting SC-ARENA with existing evaluation frameworks. We explicitly highlighted our unique contributions: **the unified "Virtual Cell" paradigm** (vs. static representations or single-task agents), **open-ended task formats** (vs. constrained multiple-choice selection), and **knowledge-grounded metrics** (vs. surface-level lexical or string matching).
* To further address comparison concerns, we evaluated the reasoning model **Cell-o1** across all five tasks and added these results to **Table 3**.

**2. Addressing Dataset Leakage Concerns (Addressing Reviewer 2kiV)**
* We added a comprehensive analysis in **Section 4.1** and **Appendix A.6** to directly address leakage risks. This revision details how our strategies, **novel task formulations**, **knowledge-grounded assessment**, **randomized sampling**, and **forward-looking benchmark design**, collectively mitigate leakage risks.
* We further supported this by adding **Table 9**, which shows that C2S models achieve low BLEU scores despite high accuracy, proving their performance stems from acquired biological knowledge rather than memorization of training samples.

**3. Validation of Evaluator Reliability and Robustness (Addressing Reviewers 2kiV, 6o9F, 6ZRo)**
* **Cross-Task Reliability:** We revised **Section 5.1** and **Appendix A.7** to include new correlation analyses. These results demonstrate that our evaluator scores align strongly with objective biological metrics (e.g., marker gene recall, DEG cosine similarity) and expert human preferences (Spearman $\rho > 0.8$) across all tasks.
* **Robustness Checks:** We added **Appendix A.8.3**, which details robustness checks where we re-ran benchmarks with a different judge model (DeepSeek-V3.2) and different random seeds. This revision confirms the evaluator's high consistency and proves it is insensitive to response length.
* **Interpretability:** In response to Reviewer 6ZRo, we added a detailed step-by-step walkthrough of the scoring mechanism in **Appendix A.9**.

**4. Standardization and Fairness of our Unified Evaluation Protocol (Addressing Reviewer 2kiV)**
* We revised **Section 4.3** and **Appendix A.5** to explicitly clarify our unified evaluation protocol. We detailed that all model outputs are converted to a standardized schema before evaluation to ensure fairness, while strictly maintaining the original inference protocols for domain-specialized models to avoid bias.

### Part II: Clarifications & Additional Textual Revisions

**5. Revisions on Evaluation Scope and Future Directions (Addressing Reviewers 2kiV, 6o9F)**
* We revised **Section 3.3** to clarify that our evaluation is grounded in established experimental truth to assess fidelity, rather than focused on novel hypothesis generation.
* We also expanded **Section 6.3** to discuss the framework's extensibility to emerging modalities, explicitly covering plans for spatial transcriptomics and developmental trajectory tasks.

**6. Clarifying Theoretical Contributions (Addressing Reviewer k5M8)**
* We revised **Section 6** to clarify that our modeling and scoring insights **are fully aligned** with the proposed paradigm, serving as **core theoretical contributions** and the **first step** in addressing the "fluent but not faithful" gap.

We believe these revisions and additional analyses substantially strengthen the manuscript and comprehensively address the reviewers' concerns. We hope this summary assists in your final assessment.

Best regards,

The Authors

---

### Meta-Review · Area_Chair_x66i · 2026-01-07

**Summary:**

Overall, the reviewers commend the proposed "virtual cell" approach for a holistic evaluation of LLMs on single-cell biology tasks.
Furthermore, the use of SC-Arena for comprehensive evaluation of latest LLMs on multiple tasks demonstrate the potential value of the constructed benchmark.
Concerns include the clarification of the novelty and main value position of the benchmark and the evaluation framework compared to prior benchmarks that exist in the field, the need for additional evaluations on existing methods on additional tasks, further rationale for model selection and additional discussion regarding the fairness of the comparisons, potential limitation due to the reliance on LLM-as-a-judge, and potential risk of data leakage.

**Reviewer Concerns:**

The authors have provided point-by-point-response to the reviewers' concerns.
This includes clarification regarding the main novelty and merits of the new benchmark, additional performance evaluation of a reasoning model, provision of detailed examples and clarifications, arguments regarding potential dataset leakage risks ad the use of LLM-as-a-judge.
During the rebuttal and discussion period, it appears that most minor concerns and several major concerns have been addressed to some extent.

**Reviewer Scores:**

Concerns raised by reviewer k5M8, who was initially most critical regarding the potential novelty & value proposition of SC-Arena, seem to have been addressed, at least partially, and score increase from the initial score of 2 to 4 (marginally below acceptance) might have been likely.

Reviewer 6ZRo was highly positive about the work, and the the additional clarifications would have resulted in maintaining the original score.

Reviewer 2kiV gave an initial score of 4, with several detailed and constructive feedback.
The authors have thoroughly addressed many of reviewer 2kiV's initial concerns through additional experiments, clarifications, and justifications.
Based on this, I expect that the reviewer might have increased the score from 4 to 5~6.

Finally, reviewer 6o9F was fairly positive about the original manuscript, giving a score of 6.
The authors response answers concerns regarding the potential impact/bias of LLM-as-a-judge approach and concerns about hallucination.
These brief answers might not fully address the reviewer's concerns, but would have been sufficient to maintain the original score.

---

### Decision · Program_Chairs · 2026-01-26

Accept (Poster)